# Cloud albedo changes in response to anthropogenic sulfate and non-sulfate aerosol forcings in CMIP5 models

Lena Frey[1], Frida A.-M. Bender[1], and Gunilla Svensson[1]

[1]Department of Meteorology and Bolin Centre for Climate Research, Stockholm University, Stockholm, Sweden

*Correspondence to:* Lena Frey (lena.frey@misu.su.se)

**Abstract.** The effects of different aerosol types on cloud albedo are analyzed using the linear relation between total albedo and cloud fraction found on monthly mean scale in regions of subtropical marine stratocumulus clouds, and the influence of simulated aerosol variations on this relation. Model experiments from the Coupled Model Intercomparison Project phase 5 (CMIP5) are used to separately study the responses to increases in sulfate, non-sulfate and all anthropogenic aerosols. A cloud brightening on month-to-month scale due to variability in the background aerosol is found to dominate even in the cases where anthropogenic aerosols are added. The aerosol composition is of importance for this cloud brightening, that is thereby region dependent. There is indication that absorbing aerosols to some extent counteract the cloud brightening, but scene darkening with increasing aerosol burden is generally not supported, even in regions where absorbing aerosols dominate. Month-to-month cloud albedo variability also confirms the importance of liquid water content for cloud albedo. Regional, monthly mean cloud albedo is found to increase with the addition of anthropogenic aerosols, and more so with sulfate than non-sulfate. Changes in cloud albedo between experiments are related to changes in cloud water content as well as droplet size distribution changes, so that models with large increases in liquid water path and/or cloud droplet number show large cloud albedo increases with increasing aerosol. However, no clear relation between model sensitivities to aerosol variations on the month-to-month scale and changes in cloud albedo due to changed aerosol burden is found.

## 1 Introduction

Aerosol particles have an impact on the radiation budget of the Earth, directly through interaction with radiation and indirectly via interaction with clouds. Taking into account the counteracting effects of scattering and absorbing aerosols, the net forcing from all aerosols, including their interactions with clouds, is estimated to be negative, implying a cooling of the climate. The magnitude of the cooling remains uncertain (Boucher et al., 2013) and although smaller forcings and narrower uncertainty ranges have been suggested (Stevens, 2015), the most recent report from the Intergovernmental Panel on Climate Change (IPCC) estimates the effective radiative forcing due to aerosols, including cloud adjustments, to -0.9 $\text{Wm}^{-2}$, with a 90% uncertainty range of -1.9 to -0.1 $\text{Wm}^{-2}$. Hereby, aerosols constitute the largest uncertainty to the total radiative forcing estimate from pre-industrial time (Myhre et al., 2013b).

One factor contributing to this uncertainty is the dependence of aerosol-cloud interactions on aerosol type. Aerosols can serve as cloud condensation nuclei (CCN) dependent on their chemical and physical properties, like hygroscopicity and size, and be

activated to form cloud droplets. Assuming a constant liquid water path (LWP), a cloud with a larger number of available CCN will have more numerous and smaller cloud droplets, and therefore a higher cloud albedo. This cloud brightening effect due to aerosols is known as the cloud albedo or Twomey effect (Twomey, 1977). Smaller cloud droplets can also increase the lifetime of a cloud by reducing the precipitation efficiency as described by the cloud lifetime or Albrecht effect (Albrecht, 1989).

Aerosols that are not efficient as CCN contribute less to cloud brightening, but may still affect cloud properties. Absorbing aerosols can cause local heating and a reduction of cloud cover, as suggested by Ackerman et al. (2000) or reduced turbulence and entrainment and an increase in cloudiness, as suggested by Wilcox (2010) and Wilcox et al. (2016). These, and the many additional possible pathways for aerosol influence on cloud properties, are difficult to disentangle, but the relative strength of the individual processes and their net effect are dependent on the properties of the underlying aerosol distribution.

In this study we separate the effects of sulfate and non-sulfate aerosols on cloud albedo. We focus on marine subtropical stratocumulus clouds, that have been found to be highly sensitive to aerosol perturbations (Wood, 2012), and coincide with regions of maximum forcing from the cloud albedo effect (Carslaw et al., 2013). Stratocumulus clouds cover more than 20% of the Earth's surface (Wood, 2012), and low clouds play a major role for the radiation budget of the Earth (Slingo, 1990; Hartmann et al., 1992) due to their reflection of shortwave solar radiation, particularly in marine subtropical regions with high 15 insolation and dark underlying surface. In addition, marine stratocumulus clouds are a main source of uncertainty concerning tropical cloud feedbacks, and their representation in climate models has been pointed out as problematic (Bony and Dufresne, 2005; Bender et al., 2006; Karlsson et al., 2008). A specific issue is the compensation between amount and brightness in low altitude and low latitude clouds, referred to as the "too few, too bright" problem, described by Nam et al. (2012), leading to an overestimation of the cloud albedo especially in tropical and subtropical regions (Karlsson et al., 2008; Myhre et al., 2013a).

A method for quantifying cloud albedo on monthly mean regional scale in climate models and satellite observations was introduced by Bender et al. (2011). Hereby, the cloud albedo was determined based on the near-linear relation between cloud fraction and albedo found in five regions of low-altitude subtropical marine stratocumulus clouds, defined in Klein and Hartmann (1993). By separation of the clear sky albedo $\alpha_{clear}$ and the cloud albedo $\alpha_{cloud}$, the total albedo $\alpha$ can be defined as

$$\alpha = \alpha_{cloud} \cdot f_c + \alpha_{clear} \cdot (1 - f_c) \tag{1}$$

with the cloud fraction $f_c$. For a given $\alpha_{clear}$, the linear relation between albedo and cloud fraction thus indicates a constant cloud albedo, which can be estimated as the sum of slope and intercept found from a linear regression of $\alpha$ onto $f_c$;

$$\alpha_{cloud} = d\alpha/df_c + \alpha_{clear} \tag{2}$$

The utility of this method has been supported by a comparison between preindustrial and present-day climate model simu-30 lations, showing a cloud albedo increase due to enhanced anthropogenic aerosol emissions (Engström et al., 2014).

Variations around the linear relation in a given climate or simulation scenario indicate additional variability in cloud albedo at a given cloud fraction, that may be related to variations in LWP or in cloud droplet number concentration (CDNC) via

aerosol, as according to Twomey (1977). The impact of aerosols on cloud albedo was investigated by Bender et al. (2016) by analysing the distribution of Aerosol Optical Depth (AOD) anomaly in the albedo-cloud fraction space defined by Eq. 1, assuming that an increase in aerosol emissions could cause an increase both in AOD and in number of available CCN, that may in turn increase the CDNC and the cloud albedo. Climate model simulations of present-day conditions showed such a cloud
brightening related to AOD-increases in all the subtropical stratocumulus regions studied, whereas no effect or in some regions a reversed relation was found for satellite data (Bender et al., 2016).

One possible explanation for these discrepancies is that the link between AOD and CDNC and thereby cloud albedo in models might be stronger than observed. The AOD is a measure of the amount of solar radiation which is reflected and absorbed by aerosols, and as such includes contributions from both CCN and non-CCN aerosols. Although AOD is correlated
with the number concentration of CCN over large spatial scales (Andreae, 2009), the AOD is hence not an aerosol metric that is directly related to cloud albedo, in the way CDNC is. For instance, dark absorbing aerosols overlying the clouds could cause positive AOD anomalies but darken the scene, i.e. decrease the albedo, in a way that is not well represented in the models. Model evaluations of the vertical distribution of aerosols have indicated that aerosol amount and absorption above clouds, and consequently atmospheric heating, is often underestimated in models (Peers et al., 2016; Myhre and Samset, 2015), and
Kipling et al. (2016) point at several factors including convective transport, emission height, vertical mixing and deposition processes to which a global model's vertical distribution of aerosol may be sensitive.

In this study we investigate in more detail how different aerosol types affect the cloud albedo and scene albedo in climate models. We use model output of sensitivity experiments from the Coupled Model Intercomparison Project phase 5 (CMIP5, Taylor et al. (2012)) with separated aerosol forcings to analyze the effect of all anthropogenic aerosols as well as sulfate and
non-sulfate aerosols, respectively. Following Bender et al. (2016) we analyze the AOD anomaly in the albedo-cloud fraction space and study the spread around the approximately linear relation to determine the influence of aerosol on albedo for a given cloud fraction, for each forcing scenario. We refer to this as "cloud brightening on month-to-month scale" and the results of this analysis are presented in Sect. 3.3. In the same section we also compare estimated cloud albedo changes between the different forcing scenarios and investigate the relation between the month-to-month variations and forced variations in cloud
albedo, as well as other potential factors determining the magnitude of the cloud albedo change induced by the aerosol forcing. Acknowledging the importance of droplet number in determining the cloud albedo, we also in a similar manner investigate the variations in vertically integrated cloud droplet number concentration (CDN) within and between the experiments. In addition, we study the representation of aerosols in the different CMIP5 models, further examining the large variation in AOD and CDN found on the global mean scale among climate models (Sect. 3.2). In Sect. 2 we describe the model output and analysis method
and we discuss and summarize our findings in Sect. 4.

## 2   Model output and data processing

To analyze the effects of separated aerosol types on the cloud albedo, we use model output from three experiments (*sstClim*, *sstClimAerosol* and *sstClimSulfate* according to the CMIP5 protocol) with different aerosol emissions. For all experiments,

the models were run in AMIP-type configuration with a fixed SST and sea ice climatology. In the reference simulation, the aerosol emissions were kept at a preindustrial level corresponding to the year 1850. We refer to this simulation as the 'Control' experiment. For the sensitivity experiments, all forcings were kept on a preindustrial level except the aerosol forcing, so that changes in the cloud albedo are assumed to be caused only by aerosol concentration changes. For the first sensitivity experiment, referred to as the 'All Aerosol' experiment, the emissions of all anthropogenic aerosol types were set to the level of year 2000 from the corresponding historical simulation. In the second sensitivity experiment, referred to as 'Sulfate Only', only the emissions of sulfate aerosols were set to the level of year 2000. The effects of non-sulfate aerosols are derived by calculating the difference between the All Aerosol and Sulfate Only simulations, cf. Zelinka et al. (2014). Adding this residual deviation to the control simulation gives the influence from non-sulfate aerosols, which we refer to as the 'Non Sulfate' case. A complementing experiment with only black carbon (BC) aerosol emissions set to the level of year 2000, will be discussed briefly as it was only performed with one model (NorESM). This experiment is referred to as 'BC Only'. With this simulation the influence of BC and other non-sulfate aerosols can be separated.

Aerosol particles which are represented in the studied models are dust, sea salt, sulfate, BC and organic matter (OM), where OM includes primary and secondary organic aerosol except for HadGEM (secondary only). All aerosol types on the preindustrial level are present in all experiments, and only anthropogenic aerosol emissions are altered in the sensitivity experiments. This means that the non-sulfate aerosol difference is assumed to be attributable to BC and OM, since these are the non-sulfate aerosols that are influenced by changed anthropogenic emissions. The aerosol emission data used in the models are described in Lamarque et al. (2010). No models used in this study include parameterizations for nitrate aerosols, although studies have shown that the global mean AOD is more precise when nitrate and secondary organic aerosols (SOA) are included (Shindell et al., 2013).

Sulfate and sea salt particles are reflecting and hygroscopic whereas carbonaceous aerosols are generally absorbing and non-hygroscopic. All models used in this study include parameterizations for the cloud albedo effect (Ekman, 2014; Takemura et al., 2005). Modifications of the cloud albedo are implemented through changes of CDNC and the effective radius. Parameterizations of CDNC differ widely, from a simplified scheme with a log-linear relationship between aerosol concentration and CDNC, e.g. Quaas and Boucher (2005), or with higher complexity where CDNC is dependent on aerosol concentration, size, composition and supersaturation, e.g. Abdul-Razzak and Ghan (2000). CDNC is in all models based on the aerosol number, except for IPSL which uses the aerosol mass. Aerosol types which contribute to the CDNC and thereby affect the cloud albedo differ between the models. Sulfate aerosols contribute in all models to the CDNC and the main contribution to cloud albedo variations within and between the experiments is expected to be caused by sulfate aerosols, also consistent with earlier studies which have shown that sulfate loading is the main contributor to CDNC (Wilcox et al., 2015; McCoy et al., 2017). Some models (CSIRO and HadGEM) also consider sea salt contributions whereas other models (CSIRO and IPSL) account for hydrophilic OM. Three models use a more complex approach where aerosols can be internally mixed and become soluble (MIROC, MRI and NorESM). The parameterization for the effective radius is dependent on CDNC and LWP and differs as well among the models and has been pointed out to cause model diversity (Wilcox et al., 2015). The cloud lifetime effect is included in all models, except IPSL (see Table 1).

Six GCMs in the CMIP5 archive provided the required output for all three experiments (see Table 1), giving the total cloud fraction, top of the atmosphere upwelling and downwelling shortwave radiation fluxes and AOD as monthly means. CDNC at cloud top is available only for three of these models, and we therefore investigate the vertically integrated quantity CDN, that is available for all models except IPSL. We study 30 years of output from each simulation. Our analysis focuses on five regions of low marine stratocumulus clouds, following Klein and Hartmann (1993); Australian (25-35°S, 95-105°E), Californian (20-30° N, 120-130°W), Canarian (15-25°N, 25-35°W), Namibian (10-20°S, 0-10°E) and Peruvian (10-20°S, 80-90°W). The total albedo is calculated as the upwelling divided by the downwelling shortwave radiation flux at the top of the atmosphere. Following Bender et al. (2016), the model output is de-seasonalized and also de-regionalized, i.e. we study deviations from the mean annual signature and the mean geographical signature in each region. This is done to capture aerosol-related variations in cloud albedo independent of time and location, relative to any large-scale seasonal or geographical patterns, such as for instance off-shore gradients of aerosol and cloud in the Peruvian region (Wyant et al., 2015). In agreement with Bender et al. (2016) who found that a correlation between AOD and cloud fraction masked the cloud brightening signal in satellite observations, the AOD anomaly is used in Sect. 3.3, instead of the absolute AOD. To obtain the AOD anomaly, the mean is subtracted from the AOD for each given cloud fraction. CDN is processed in the same way, for consistency.

## 3    Results and discussion

### 3.1    Global distribution of aerosol changes

In the present study, the All Aerosol experiment, where all aerosols are at a level corresponding to year 2000, is taken as a representation of the present-day aerosol distribution. Four of the models utilized (CSIRO, HadGEM, IPSL, MIROC) are also included in a previous model-intercomparison study performed by Shindell et al. (2013) within the ACCMIP (Atmospheric Chemistry and Climate Model Intercomparison Project, Lamarque et al. (2013)), who found a general agreement with observations in terms of total AOD, but pointed at model underestimates particularly over East Asia and Europe, and large inter-model differences. Examination of the All Aerosol total AOD (not shown) confirms these results and in addition, it is found that NorESM underestimates the AOD especially in South Asia (also shown by Kirkevåg et al. (2013)), and that MRI (Yukimoto et al., 2012) underestimates the global total AOD (not shown).

Although the same anthropogenic aerosol emissions by mass are used for all models, the AOD varies among models. Model diversity in AOD is caused by model differences in aerosol number and concentration and the parameterization of radiative properties of aerosol types. The aerosol loading (the column integrated aerosol mass in $\mathrm{kg\,m^{-2}}$) differs widely between models; for instance sulfate aerosol mass shows a spread of a factor of 4 in the global mean (Wilcox et al., 2015).

To map changes in AOD due to different types of anthropogenic aerosols we analyze the spatial distribution of the 550 nm AOD differences between the Control and sensitivity experiments described in Sect. 2. Figure 1 shows relative changes due to changes from preindustrial to present-day levels of all anthropogenic aerosols, sulfate and non-sulfate aerosols, respectively, for all models. A general increase in total AOD due to the changes in aerosol emissions can be seen for the All Aerosol experiment (Figure 1a), with a pattern representing the combination of those from sulfate and non-sulfate aerosol increases (Figures 1b

and c). Due to short aerosol lifetime, the distribution of AOD differences largely reflect the emission sources of anthropogenic aerosols. The main biomass burning regions cause increases in non-sulfate aerosols over central Africa, South America and Southeast Asia. Industrial pollution by sulfate aerosols is high over Europe, North America and Southeast Asia. The regions of stratocumulus cloud maxima, marked with boxes in Fig. 1, in most cases do not overlap with the maximum changes in AOD, but for the All Aerosol case (Figure 1a) significant increases in AOD are seen in all studied regions and models, except for the Canarian region in MIROC due to a reduction in dust loading of around 30%, and the Australian region in HadGEM, MIROC and MRI. The focus regions in the southern hemisphere are in general more influenced by non-sulfate aerosols while the regions in the northern hemisphere are dominated by sulfate aerosols, in agreement with previous studies (e.g. Ramanathan et al. (2001)). For NorESM, the BC Only experiment indicates that increases in AOD due to BC are comparatively small, and in some cases counteracted by decreases in OM, particularly in the Californian region, as seen from the difference between Figures 1c and d.

IPSL shows the largest AOD change in all regions, whereas HadGEM shows the smallest changes. MIROC shows negative deviations in the Canarian region, but without statistical significance, consistent with high variability of background aerosol in that region. Decreased AOD in large parts of North America in the Non Sulfate case are related to decreases in OM emissions, cf. Bond et al. (2007); Lamarque et al. (2010).

The distribution of CDN-changes (not shown) broadly coincides with the distribution of AOD-changes. Increases in CDN are primarily related to an increase in sulfate loading, as expected from the model parameterization of CDNC and in accordance with Wilcox et al. (2015); McCoy et al. (2017). But also non-sulfate aerosols cause a CDN increase, especially in the southern hemisphere. In agreement with Quaas et al. (2009) who studied relations between CDNC and AOD, HadGEM shows large increases in CDN, compared to the other models.

## 3.2 AOD and CDN Variability

The absolute AOD, and its variability varies among regions and models, as seen in Figure 2.

The year 2000 emissions, used in the All Aerosol experiments, are expected to be representative for the period 2002-2015 and model results can thus be compared with MODIS satellite observations. Fig. 2 shows that CSIRO overestimates and MIROC and MRI underestimate the AOD compared to observations in all regions. HadGEM, IPSL and NorESM are generally in closest agreement with the observations, although HadGEM overestimates AOD and variability in the Australian region, consistent with an overestimation of dust loading in this area (Bellouin et al., 2011).

MIROC and MRI show the best agreement with observations in terms of variability, whereas the other models overestimate the variability. For the Canarian region, all models show too large variability compared to observations. The variablity is overall largest for CSIRO, but HadGEM shows a large variability in the Australian region, as noted above.

These results are overall consistent with previous study results on the global scale, where HadGEM agrees well with observations, IPSL and MIROC underestimate the mean AOD and CSIRO overestimates the AOD compared to observations (Shindell et al., 2013).

The median AOD increase is highest due to an increase of all anthropegnic aerosols and shows the strongest response in the Californian region, with for example 65% increase for the model CSIRO. The Control experiment shows a similar variability as the All Aerosol experiment in all regions, indicating a high variability of the preindustrial background aerosol and only a slight increase in variability caused by added anthropogenic aerosols. Exceptions are found particularly in the Namibian region,

where the models CSIRO and HadGEM as examples show increases by 55% and 53% respectively. The variability in the Non Sulfate case is typically larger than that of the Sulfate Only case, indicating that the added variability from anthropogenic aerosol increases comes primarily from non-sulfate aerosols. De-seasonalizing the time series (as described in 2) has little effect on the variability range, i.e. the seasonal pattern of variation is not a main contributor to the AOD variability in these regions (not shown).

We note that the regional mean change in AOD between experiments, i.e. due to the addition of anthropogenic aerosol emissions, is typically small compared to the total spatio-temporal variability in AOD within the individual experiments, shown in Fig. 2. For the Canarian region the background variability greatly exceeds the AOD difference between experiments for all models, whereas for the other regions the AOD change is of similar magnitude as the interquartile range in AOD in the models with smallest variability.

Analysis of the loading of individual aerosol types can give an indication of the aerosol types dominating the variability in total AOD in the different regions. The relative aerosol loading contributions for all models for the different regions in the control simulation are listed in Table 2. The mass loading in all regions is dominated by dust and sea salt, consistent with these aerosol types forming primarily large and heavy particles. In the Canarian region, the dust loading is between 70 and 90 % of the total aerosol loading. The median AOD and AOD variability is considerably higher in the Canarian region compared

to the other regions. The BC mass loading is negligible compared to other aerosol types for all regions and models. NorESM consistently shows a larger fraction of OM than the other models, and NorESM and CSIRO in general have higher fractions of sulfate contribution to the total loading, compared to the other models. IPSL was noted to underestimate OM by 20% in the used CMIP5 simulations (Dufresne et al., 2013). The changes in loading between the experiments are however determined by the aerosol types with anthropogenic sources, as noted in Section 1.

Satellite retrievals of CDNC are not trivial and differ widely (McCoy et al., 2017; Bennartz, 2007; Zeng et al., 2014) and there is also large variation among the studied models in their estimates of CDN (not shown). CDN changes due to added anthropogenic aerosols are concurrent with AOD changes, with the highest increase in the Californian region, where the median increases by up to 48% in the CSIRO model. In contrast to the AOD variability which is driven mainly by non-sulfate aerosols, the CDN variability is primarily related to anthropogenic sulfate aerosols, consistent with CDN in the models being

a function of the total amount of hydrophilic aerosols.

### 3.3   Cloud albedo and cloud brightening effect

To illustrate the aerosol-induced cloud brightening on the month-to-month scale, we first investigate the co-variation of CDN with the albedo at a given cloud fraction, i.e. of the CDN-anomaly distribution in albedo-cloud fraction space. This is shown in Figure 3 for all experiments for NorESM in the Californian region as an example. It is clear from Figure 3 that positive

CDN anomalies are consistently related to higher albedo for a given cloud fraction, i.e. that the cloud albedo is higher at higher CDN. Consistently positive gradients in the albedo-cloud fraction space are in the same way found for LWP (not shown), in accordance with Bender et al. (2016).

In line with Bender et al. (2016) we next turn to the distribution of AOD anomaly in albedo-cloud fraction space, and find that the Californian region in NorESM shows positive gradients for all experiments (Figure 4). However, the emerging AOD anomaly gradient displays variation between experiments, models and regions, in a way that the CDN anomaly gradient does not. To quantify the direction and strength of the gradient, two separate linear regressions are performed for the points falling above the 90th and below the 10th percentile of the AOD anomaly range respectively. The two separate regression lines are indicated in Figure 4, and the difference between the derived cloud albedo for the upper and lower regression lines is referred to as the gradient strength. Figure 5 summarizes these AOD gradient strengths for all regions, all experiments and all models. Figure 3 in a similar way shows separate regression lines for "high" and "low" CDN, for the same example region and model. For CDN anomaly, all models show positive gradients for all experiments in all regions, indicating the importance of CDN to cloud albedo. CDN as a vertically integrated quantity is however not independent of cloud depth, and LWP. An estimate of CDNC at cloud top which could be used to separate the effect of droplet number from cloud thickness is only available for three of the models, and the gradients for this variable are not consistent between models. Similarly, cloud-top effective radius does not give consistent gradients. This indicates that the month-to-month cloud albedo variability is more strongly driven by LWP variations than variations in droplet size distribution.

A predominantly positive AOD gradient appears for all models in the five regions for the All Aerosol experiment, in agreement with what Bender et al. (2016) found for present-day simulations for a larger set of CMIP5 models. For the Control experiment, the cloud albedo also co-varies with aerosols and positive gradients are seen, which indicates that a cloud brightening on month-to-month scale also occurs due to preindustrial aerosols only. For the Namibian and Peruvian regions there are cases of negative gradients.

Comparing the gradient strengths (Fig. 5), there is no systematic strengthening of the AOD gradient between the Control and All Aerosol experiments, which supports the idea that month-to-month scale cloud brightening by preindustrial aerosols is similar in strength to that induced by total (preindustrial and anthropogenic) aerosols. This is also in agreement with the relatively small difference in variability between the Control and All Aerosol experiments found in Sect. 3.2. In most models, both the Sulfate Only and the Non Sulfate cases show positive gradients, again consistent with the background aerosol, rather than the added anthropogenic aerosols, being largely responsible for the gradient. For CSIRO in the Nambian region, the gradients in the Control and Sulfate Only cases are clearly positive, but in both the Non Sulfate and All Aerosol cases they are weaker, or negative. This is consistent with the increased non-sulfate aerosol contributing to the AOD and counteracting the background cloud brightening on the month-to-month scale.

The gradients are overall weakest for the Namibian region, where BC and OM are typically important for explaining the AOD variability, consistent with absorbing aerosol counteracting the cloud brightening, as suggested by Bender et al. (2016). The positive gradients are on the other hand strongest in the dust-dominated Canarian region, where satellite observations indicate negative gradients (Bender et al., 2016). The aerosol composition appears to be more important for the gradient

strength than the total AOD variability; the Canarian region displaying the strongest gradients has the largest AOD variability, but the Namibian region with the weakest gradients also typically shows higher variability than the remaining regions (see Fig. 2).

As described in Sect. 2, the cloud albedo for each model, region and experiment, can be estimated from a linear regression
for the linear relation between cloud fraction and albedo. The deviation in cloud albedo between the sensitivity experiments and Control simulation shows, that the cloud albedo is enhanced due to increased anthropogenic aerosol emissions for most of the models and regions, see Fig. 6. In agreement with Zelinka et al. (2014), both sulfate and non-sulfate aerosols enhance the cloud albedo in general, but the increase in cloud albedo is typically larger for sulfate than non-sulfate aerosols, in agreement with the larger changes in CDN in the Sulfate case (Sect. 3.2). In the Australian region several models indicate decreased cloud
albedo due to changes in non-sulfate aerosol, consistent with decreased CDN.

Previous studies have shown that aerosol forcings are not necessarily linearly additive (Stier et al., 2006; Jones et al., 2007), but here the sum of the effects of anthropogenic sulfate and non-sulfate aerosols on the cloud albedo is within the uncertainty range of the effect of all anthropogenic aerosols, in most cases.

The largest cloud albedo change is found in the Californian and Canarian regions with a relative increase of up to 8%
compared to the control experiment, in agreement with Engström et al. (2014), as opposed to the Australian region with the lowest cloud albedo changes by less than 1%. The latter region is more influenced by non-sulfate aerosols in contrast with the regions in the northern hemisphere which are more influenced by sulfate aerosols (see Sect. 3.1). The greatest model diversity in terms of cloud albedo response also occurs in the Californian and Canarian regions, whereas in the Australian, Namibian and Peruvian regions the models are in closer agreement with each other. The three models MIROC, MRI and NorESM have
similar parameterizations for the cloud albedo effect, but show a different cloud albedo response due to changed anthropogenic aerosol emissions. MIROC has small cloud albedo changes in all regions, whereas the changes for MRI and NorESM are in some cases large. Quaas et al. (2009) have suggested that HadGEM has too strong CDNC increases with increasing AOD and this model indeed shows the largest cloud albedo changes. IPSL, which is the only model that does not include a cloud lifetime effect, shows comparatively small changes in cloud albedo, consistent with Zelinka et al. (2014) who found a low
effective radiatve forcing due to aerosol-cloud interactions for this model compared to other CMIP5 models. IPSL is also the only model which prescribes CDNC based on the aerosol mass, and since microphysical processes are important for aerosol-cloud interactions, using the aerosol mass instead of aerosol number could affect the cloud albedo response in this model (cf. Gettelman (2015)).

Even though the cloud albedo changes in the five regions are small, within 0.03, the aerosol changes have a substantial local
radiative effect. The difference in upwelling shortwave raditaion between the All Aerosol and Control experiments, i.e. the total radiative effect due to changes in both cloud fraction, cloud albedo and clear-sky albedo caused by anthropogenic aerosols, are as large as -6.0 $\mathrm{Wm}^{-2}$ in the Californian region. The largest changes in radiation occur in the Californian and Canarian regions, consistent with the larger cloud albedo changes there, and MRI and HadGEM show the greatest changes in shortwave radiation, consistent with the higher cloud albedo changes in those models.

To investigate what controls the sensitivity of changes in cloud albedo in response to anthropogenic aerosol changes in this set of models, we analyze relations between cloud albedo change and potential explaining factors among the models. The correlation coefficients in Fig. 7 are based on five points only, representing all models except IPSL that does not provide CDN, but may still give an indication of the sign and strength of the relations between cloud albedo change and other factors, as follows.

As expected from the configuration of the cloud albedo effect in the models, the change in CDN is positively correlated with the cloud albedo change in all regions (correlation coefficients ranging from 0.46 to 0.93). The strongest response occurs in the Canarian region, consistent with a high cloud albedo change in this region.

It has been suggested that for a given change in sulfate loading, the magnitude of the cloud albedo effect should be larger for models with a low burden, due to cloud droplet size being more sensitive to changes in sulfate loading at low burden (Carslaw et al., 2013; Wilcox et al., 2015). The preindustrial sulfate loading shows a weak negative correlation (correlation coefficients ranging from 0.17 to 0.46) with CDN changes and cloud albedo change, except in the Californian region, i.e. models with lower preindustrial sulfate burden show a larger change in cloud albedo between the Control and All Aerosol experiments. The correlation between change in AOD and change in cloud albedo is only weakly positive for the Californian and Canarian regions and weakly negative for Australian, Namibian and Peruvian regions.

Turning to gradient strengths, all models show a positive correlation between changes in cloud albedo and the strength of the CDN anomaly gradient, i.e. models with higher sensitivity of cloud albedo to CDN on month-to-month scale also have larger cloud albedo response to aerosol increases. A region-dependence is detectable, where the weakest correlation occurs in the Californian region with a correlation coefficient of 0.09 and the strongest in the Australian region with 0.78.

The strength of the AOD anomaly gradient in contrast, is overall not well correlated with the cloud albedo change, suggesting that models with stronger cloud brightening effect on the month-to-month scale do not necessarily have a stronger cloud albedo effect due to anthropogenic aerosol changes. However, IPSL that has the weakest AOD gradient strengths overall, also shows small changes in cloud albedo, and MRI that has strong positive gradients also in general shows large differences in cloud albedo between experiments (cf. Figures 5 and 6).

As cloud albedo is strongly dependent on LWP, we also examine the change in LWP between experiments. From the experiment setup of the fixed SST simulations, only small changes in LWP are expected, due to small changes in the land temperature which can affect the circulation and thereby cloud properties (Erickson et al., 1995; Hansen et al., 2005; Allen and Sherwood, 2011; Lewinschal et al., 2013). But all models, except IPSL, include the cloud lifetime effect, i.e. higher aerosol concentrations in the sensitivity experiments leading to smaller and more cloud droplets and thereby reducing the precipitation efficiency and increasing the LWP. An increase in LWP leads to a higher cloud albedo, which can enhance the cloud brightening by the Twomey effect. Total changes in LWP between experiments (circulation and aerosol driven) are typically positive, and in most regions also positively correlated with the change in cloud albedo. For the Namibian region no signifcant correlation can be detected, but for the Australian, Californian, Canarian and Peruvian regions correlation coefficients are between 0.51 and 0.74, see Figure 7.

In summary, Figure 7 suggests that the strength of the month-to-month cloud brightening in terms of CDN variation, changes in CDN and changes in LWP are driving cloud albedo changes between the experiments. We also note that the changes in LWP between experiments make it difficult to isolate the cloud albedo effect, which in theory acts at constant LWP. All models except one include cloud lifetime effects, which have been suggested both to be uncertain, and to contribute significantly to the total aerosol-cloud interaction in models (Gettelman, 2015).

## 3.4 Vertical aerosol and cloud distributions

Absorbing aerosols overlying low clouds may cause a scene darkening. This was suggested by Bender et al. (2016) as an explanation for high AOD being related to low albedo for a given cloud fraction, as was indicated by satellite data for the Canarian and Namibian stratocumuls regions. As in the historical CMIP5 simulations discussed by Bender et al. (2016), the model simulations studied here with few exceptions display positive gradients, but it is still of interest to examine the vertical distribution of clouds and aerosols in these models, to investigate if they show any relation to the gradient direction and strength.

Vertically resolved aerosol extinction, which can be integrated over the atmospheric column to yield the total AOD, is only available for three models (MIROC, IPSL and NorESM) in the CMIP5 archive. All three models show an aerosol layer above the low clouds in the Canarian region, consistent with observations (Chand et al., 2008; Devasthale and Thomas, 2011; Waquet et al., 2013; Winker et al., 2013), with a contribution of 50% to the total AOD. Figure 8 shows vertical cross sections of the aerosol extinction coefficient at 550 nm and the vertically resolved cloud fraction for the model MIROC, in the Canarian region, as an example, and an aerosol layer co-located with the cloud layer as well as an overlying aerosol layer is evident for all four experiments. The overlying aerosols here are assumed to be dust aerosols as the dominant loading type is dust in the Canarian region. Dust aerosols are parameterized as weakly absorbing in the models and could hence reduce the scene albedo or affect the clouds by changing the heating profiles, similar to BC. MIROC however, shows positive gradients in the Canarian region (cf. Fig. 5), although somewhat weaker than for other regions, especially in the Non Sulfate case. NorESM has distinct positive gradients in the Canarian region. The strongest negative gradient (Fig. 5) is seen for MIROC in the Namibian region, where overlying aerosols are found only for parts of the year, not affecting the annual mean. On the whole, there is no indication of systematic presence of overlying aerosol in the regions where weak or negative gradients are seen. Hence, to the extent that overlying aerosols occur in these models, they are not sufficiently absorbing to overcome the cloud brightening and create negative AOD gradients, as those seen in satellite data (Bender et al., 2016), and there is no clear dependence of gradient strength on overlying aerosol.

MRI overall has strong AOD gradients and large sensitivity, and is also one of the models that includes aerosol interactions with ice clouds (Rotstayn et al., 2013). An analysis of the vertical cloud fraction shows that for MRI in the Californian region, high clouds frequently occur above the low clouds, and enhanced cloud albedo from high clouds could therefore increase the total albedo and contribute to scene brightening. However, the Californian region does not stand out as more sensitive than other regions, with less high cloud occurrence. For MIROC, that also includes aerosol interaction with ice clouds (Rotstayn et al., 2013), high clouds are seen in the Namibian region, where gradients are found to be negative. Hence, ice cloud interaction

does not seem to be important for the total effect of aerosol on cloud albedo, and the predefined regions in general contain mainly low clouds, in agreement with what is seen in satellite observations (Bender et al., 2016).

### 3.5 Black carbon influence on the cloud albedo

For NorESM an additional BC Only experiment was carried out. The largest changes in AOD due to BC aerosols can be seen in the biomass burning regions in South America, equatorial Africa and Southeast Asia, see Fig. 1. There are no significant increases in CDN due to the BC increase. The vertical distribution of BC aerosols is important for the radiative forcing in models (Samset et al., 2013) in general, and strongly absorbing BC aerosols above marine stratocumulus clouds could potentially darken the scene and lower the albedo, but as discussed in Sect. 3.4, there are no overlying aerosols above the cloud layer in the main regions of BC loading, the Namibian and Peruvian region, in the model simulations studied here.

As was found for for the other experiments, the CDN-gradient in albedo-cloud fraction space (Fig. 3) remains positive in the BC Only case, again showing that cloud albedo is largely determined by the CDN. The AOD gradient for the BC Only experiment is shown in Fig. 4 for the Californian region, and gradient strengths for all regions are summarized in Fig. 5. In the Peruvian and Namibian regions where the BC loading is high, the gradients are weaker or reversed, compared to the Non Sulfate case, where BC as well as OM is increased, indicating that the BC may actually have an effect of counteracting and dampening the cloud brightening. In the other regions the gradients remain positive, which supports the idea, that the natural background aerosol causes a cloud brightening that the additional BC can not counteract. It is noteworthy that relative AOD changes due to increased BC emissions on the global scale are tenfold smaller than due to all non-sulfate aerosols (Fig. 1), but it is also noteworthy that absorption by BC is commonly underestimated in models, (Bond et al., 2013), suggesting that the dampening of the cloud brightening could in fact be underestimated.

## 4 Conclusions

The aim of this study is to determine the effects of different aerosol types on the cloud albedo in an ensemble of climate models. Six CMIP5 models provided output from sensitivity experiments where total and sulfate-only aerosol emissions were separately changed to present-day levels, while all other anthropogenic forcings were kept at a preindustrial level. The models all parameterize the cloud albedo effect, i.e. higher aerosol concentration leading to an increase of the cloud droplet number concentration and a decrease of droplet size at constant liquid water content, with a resulting increase in cloud albedo with increasing amount of aerosol. We use the methods of Bender et al. (2011, 2016) to examine the relation between CDN, LWP, AOD and cloud albedo in five marine subtropical stratocumulus regions, separating the effects of anthropogenic sulfate and non-sulfate aerosols from the preindustrial background aerosols.

The representation of the AOD and its change from preindustrial to present-day conditions, shows large differences between the models confirming results from previous studies (Shindell et al., 2013). An increased AOD, due to changed emissions, can be identified in the five study regions for almost all models. The Canarian and Australian regions are less influenced by anthropogenic aerosols than the other regions, and particularly the Canarian region is largely dust-dominated. The Californian

region is mainly influenced by sulfate aerosols while the Namibian and Peruvian regions are dominated by non-sulfate aerosols. The addition of anthropogenic aerosols increases the AOD variability from that of the background state, and the variability increases more due to addition of non-sulfate than sulfate aerosols. The CDN variability follows the AOD response with largest changes in the Californian region, but the variability is in the case of CDN most closely related to sulfate aerosols.

For all models, in all regions and for all experiments, the CDN is closely related to the cloud albedo. Positive (negative) anomalies in CDN consistently correspond to higher (lower) albedo at a given cloud fraction, indicating an aerosol-related cloud brightening on the month-to-month scale. However, CDN is a vertically integrated quantity, and therefore not independent of cloud thickness in terms of LWP. Cloud-top estimates of CDN and effective radius were available for only a subset of the models and do not show consistent relations with cloud albedo. LWP on the other hand shows positive co-variation with cloud

albedo for all models and all regions, both directly and via the vertically integrated CDN, illustrating the importance of cloud thickness (LWP) in determining cloud albedo variability.

A general co-variation between cloud albedo and AOD is also found, represented by a positive gradient in AOD anomaly in albedo-cloud fraction space. Similar signals have been found in simulations of present-day climate (Bender et al., 2016), but here positive gradients are found not only in the presence of all anthropogenic aerosols, but also in the reference experiment

with aerosols at preindustrial level, and when sulfate and non-sulfate aerosols only are increased. Only a slight strengthening of the dependence occurs with additional anthropogenic aerosols compared to the case with preindustrial conditions, which supports the notion of a cloud brightening effect by background aerosols. Previous studies, e.g. Carslaw et al. (2013) have also shown that the albedo sensitivity to CCN is higher in clean preindustrial conditions with low CDNC compared to present-day conditions.

The month-to-month scale cloud brightening as given by the AOD anomaly gradient is generally greatest for the dust-dominated Canarian region, whereas the weakest cloud brightening signals and the most instances of reversal of the AOD anomaly gradient, indicating aerosol induced scene darkening, are seen in the more BC-dominated Namibian and Peruvian regions, particularly in an additional experiment isolating BC aerosol increases.

The aerosol composition is hence of importance for the gradient strength, as is to be expected given the aerosol species

that are parameterized to contribute to CDNC in the models. However, the cloud brightening shows no clear dependence on the vertical distribution of the aerosols. Dust aerosols above the clouds are found in the Canarian region, but BC aerosols are not found to be prevalent above the clouds in the biomass burning regions, which may be expected from observations (Chand et al., 2008; Devasthale and Thomas, 2011; Waquet et al., 2013; Winker et al., 2013), and which may further counteract the cloud brightening. Hence, the darkening effect of the anthropogenic non-sulfate aerosols is not strong enough in the models to

counteract the brightening effect of the background aerosol, leaving climate models with predominantly positive gradients in disagreement with satellite observations (Bender et al., 2016).

However, the strength of the month-to-month cloud brightening in terms of AOD variation does not seem to be critical for the magnitude of the cloud albedo change between experiments. The regional mean cloud albedo is estimated for the different forcing scenarios, based on the near-linear relation between albedo and cloud fraction. An enhanced cloud albedo due

to addition of both anthropogenic sulfate and non-sulfate aerosols is found, consistent with Zelinka et al. (2014), but sulfate

aerosols tend to cause a larger increase in cloud albedo than non-sulfate aerosols, consistent with the importance of sulfate aerosols to CDNC and the cloud albedo dependence on CDNC. The sum of the cloud albedo changes due to sulfate and non-sulfate aerosols are within the uncertainty ranges of the changes due to all aerosols. The cloud albedo increases are typically larger for the regions in the northern hemisphere, consistent with the dominance of sulfate aerosols in these regions. We note however, that the change in AOD between experiments is typically small compared to the AOD variability within experiments; this is true for all models in the Canarian region, and for most models in the remaining four regions. Models with stronger AOD variation on the month-to-month scale do not show systematically higher cloud albedo change with addition of anthropogenic aerosols. This means that the discrepancy between models and observations in terms of cloud brightening on month-to-month scale does not necessarily have consequences for the effective radiative forcing due to aerosol-cloud interactions. The month-to-month variation in CDN is more important for cloud albedo changes, so that models with stronger CDN anomaly gradient tend to show larger cloud albedo change between Control and All Aerosol experiments.

The spread in cloud albedo change in this ensemble of models was found to be related to differences in CDN changes between the experiments and hence the parameterizations of the CDNC, which has been found important also in previous studies (Penner et al., 2006; Storelvmo et al., 2009; Wilcox et al., 2015). Among models with a prognostic CDNC scheme (NorESM, MIROC, MRI) some are more and some less sensitive to aerosol changes compared to models with a diagnostic scheme. Similarly, the models MIROC and MRI that include aerosol interaction with ice clouds (Rotstayn et al., 2013) can not be singled out in terms of cloud albedo sensitivity in these stratocumulus-dominated regions, although ice cloud effects have been found to be important for radiative forcing due to aerosol-cloud interaction on global scale (Zelinka et al., 2014).

Finally, the change in regional mean cloud albedo induced by the addition of anthropogenic aerosols seems to be largely driven by changes in LWP between the experiments. These changes may be induced by aerosol interaction with the clouds through the cloud lifetime effect, by which more aerosol particles leads to smaller droplets and reduced precipitation efficiency, or by aerosol forcing causing changes in circulation that affect the cloud properties indirectly (Erickson et al., 1995; Hansen et al., 2005; Allen and Sherwood, 2011; Lewinschal et al., 2013). According to Gettelman (2015), cloud lifetime effects contribute one-third to simulated aerosol-cloud interactions, and George and Wood (2010) found variability in cloud radiative properties to be dominated by cloud cover and liquid water path rather than microphysics. This elucidates the difficulty of isolating aerosol effects on clouds from meteorological variations (Stevens and Feingold, 2009; Engström and Ekman, 2013; Peters et al., 2014; Rosenfeld et al., 2014; Feingold et al., 2016) not only in observations, but also in models.

## 5   Code availability

Code is available from the corresponding author upon request.

## 6   Data availability

The CMIP5 data is available through https://esgf-data.dkrz.de/projects/esgf-dkrz/.

*Competing interests.* The authors declare that they have no conflict of interest.

*Acknowledgements.* We would like to thank two anonymous reviewers for their comments which improved our article. We acknowledge the World Climate Research Programme's Working Group on Coupled Modelling, which is responsible for CMIP, and we thank the climate modeling groups (listed in Table 1 of this paper) for producing and making available their model output. For CMIP the U.S. Department of Energy's Program for Climate Model Diagnosis and Intercomparison provides coordinating support and led development of software infrastructure in partnership with the Global Organization for Earth System Science Portals. The post-processing and analysis of model output were performed on resources provided by the Swedish National Infrastructure for Computing (SNIC) at the National Supercomputer Centre at Linköping University (NSC). The authors thank the Earth System Grid Federation (ESGF) and Centre for Environmental Data Archival (CEDA) for providing the CMIP5 data archive.

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

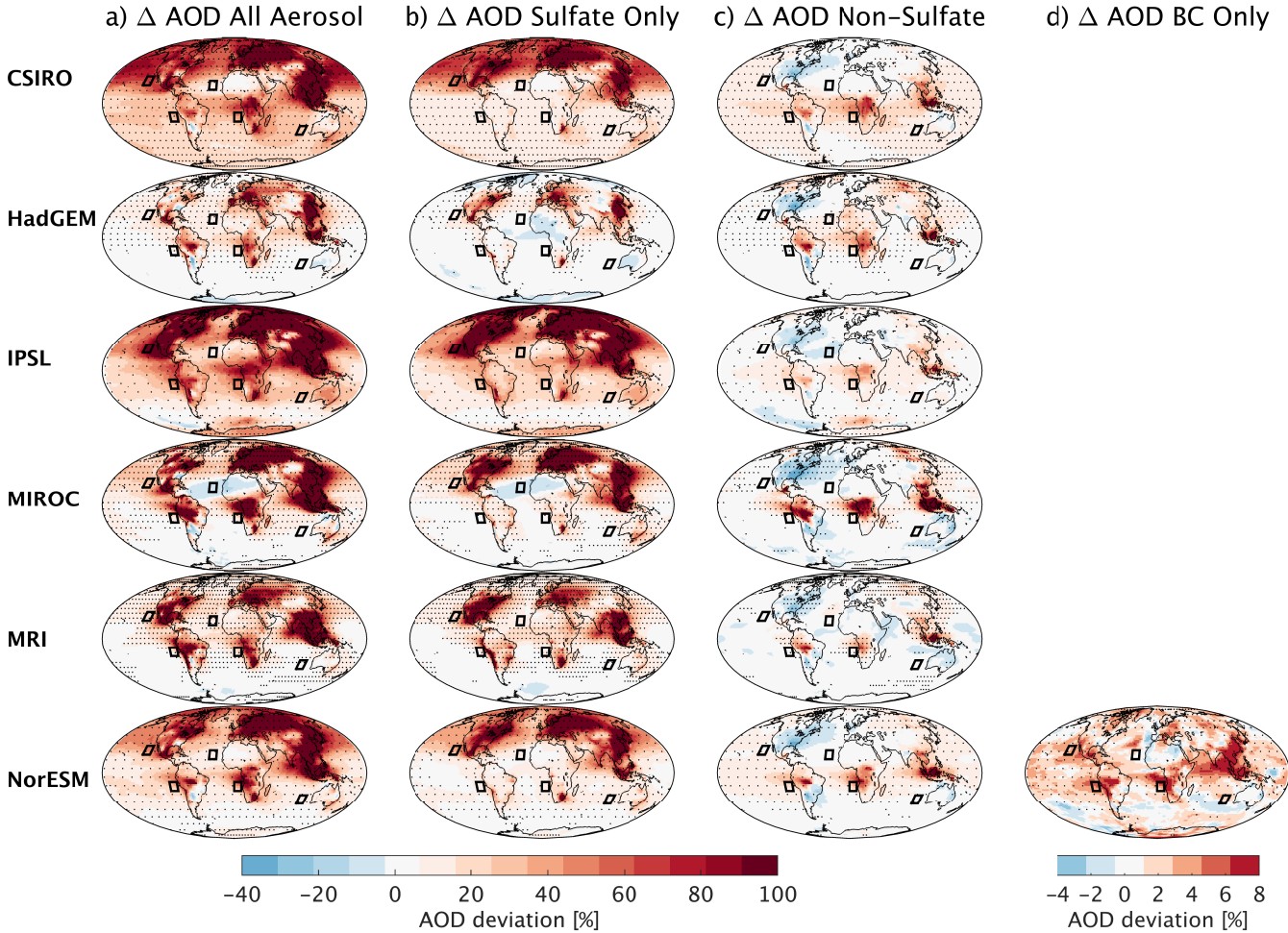

**Figure 1.** Relative AOD deviation in % between a) the All Aerosol and Control experiment, b) the Sulfate Only and Control experiment, c) the All Aerosol and Sulfate Only experiment for six CMIP5 models, and d) the BC Only and Control experiment for one model. Statistical significance at the 5% level, determined with a t-test, is indicated with stippling, interpolated to a coarser grid. Black boxes indicate the five analysis regions of marine stratocumulus clouds.

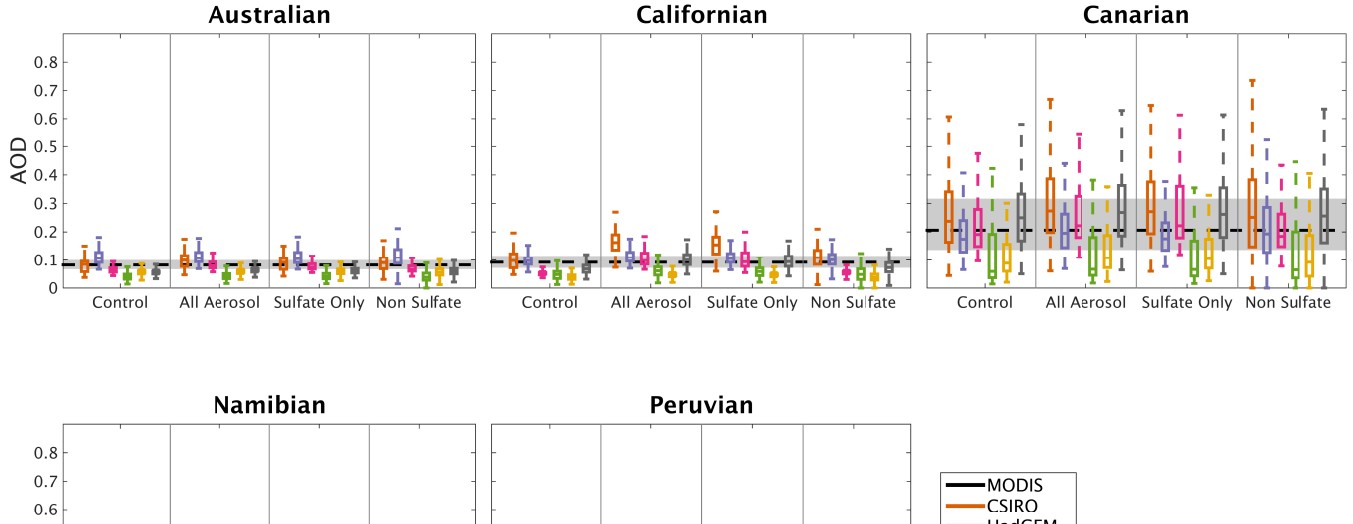

**Figure 2.** Box-and-whisker plots of total AOD for six CMIP5 models for the Control, All Aerosol, Sulfate Only and Non-Sulfate cases in the Australian, Californian, Canarian, Namibian and Peruvian regions. Median, 25th and 75th percentiles and maximum and minimum AOD are indicated. The data are not de-seasonalized and not de-regionalized. The black dashed line indicates the median AOD value for MODIS satellite observations from 2002 to 2015, and grey shading shows the range between the 25th and 75th percentiles. Changes between the experiments are significant at the 5% level, except for two models in two different regions.

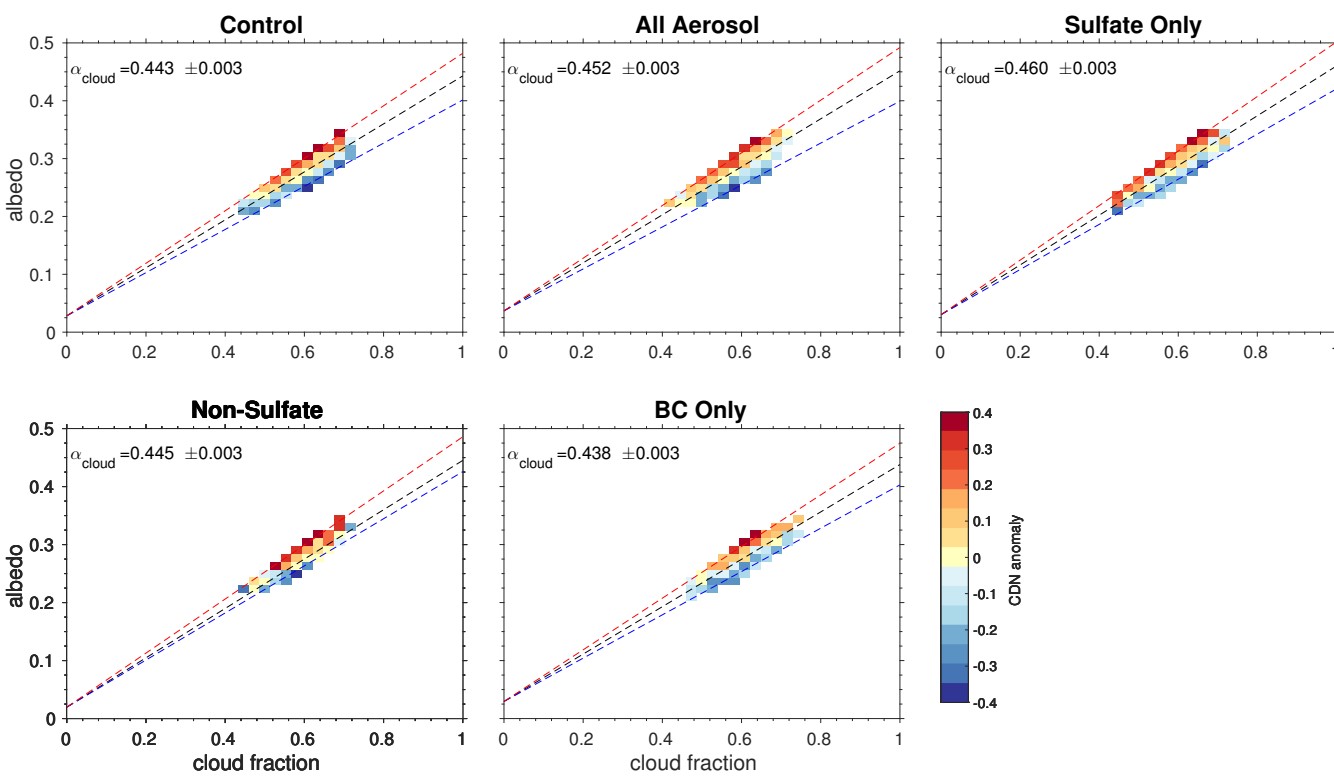

**Figure 3.** CDN anomaly gradient for the model NorESM for all experiments, in the Californian region. Two linear regressions are performed for the separated upper (red dashed line) and lower (blue dashed line) 10% of the data. The black dashed line represents a linear fit for all the CDN anomaly data, and the estimated cloud albedo is derived from the slope and intercept of that linear regression. The color scale for the anomaly was normalized by the standard deviation of CDN for each cloud fraction bin.

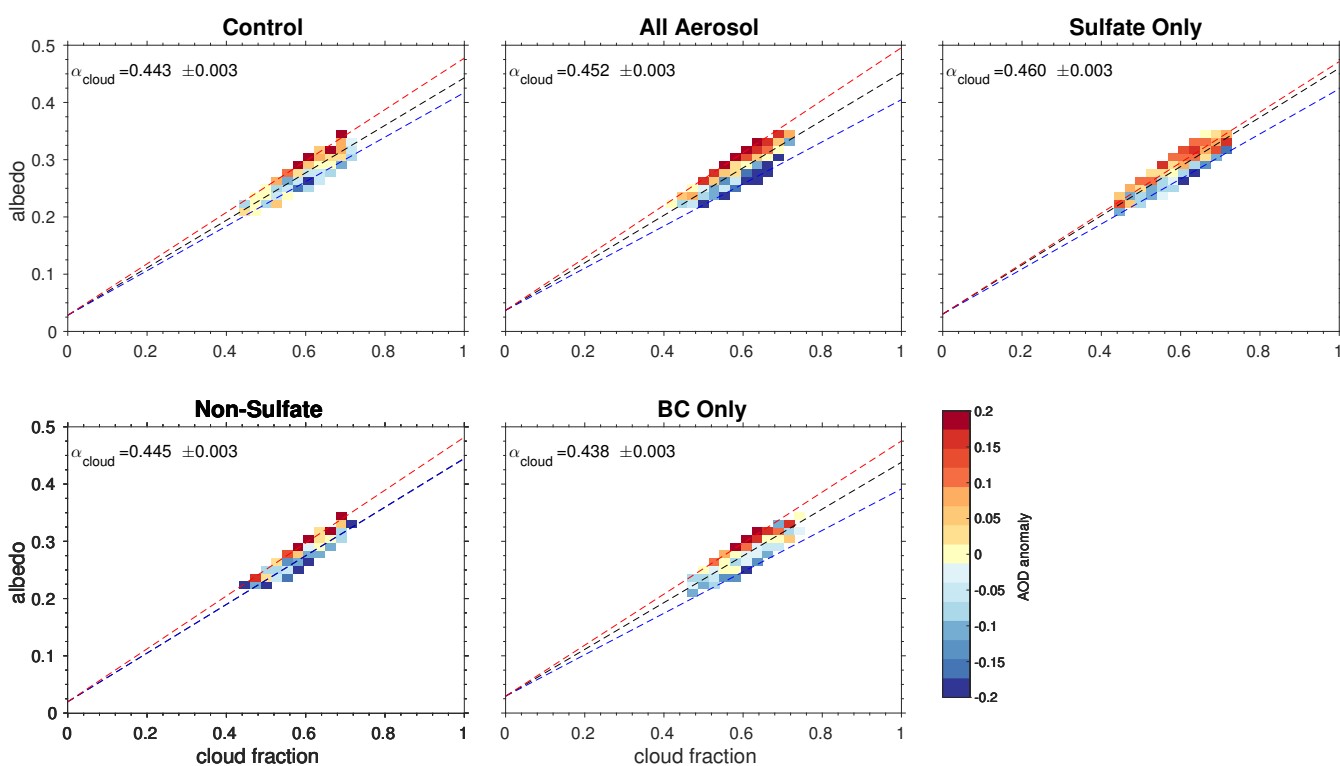

**Figure 4.** AOD anomaly gradient for the model NorESM for all experiments, in the Californian region. Two linear regressions are performed for the separated upper (red dashed line) and lower (blue dashed line) 10% of the data. The black dashed line represents a linear fit for all the AOD anomaly data, and the estimated cloud albedo is derived from the slope and intercept of that linear regression. The color scale for the anomaly was normalized by the standard deviation of AOD for each cloud fraction bin.

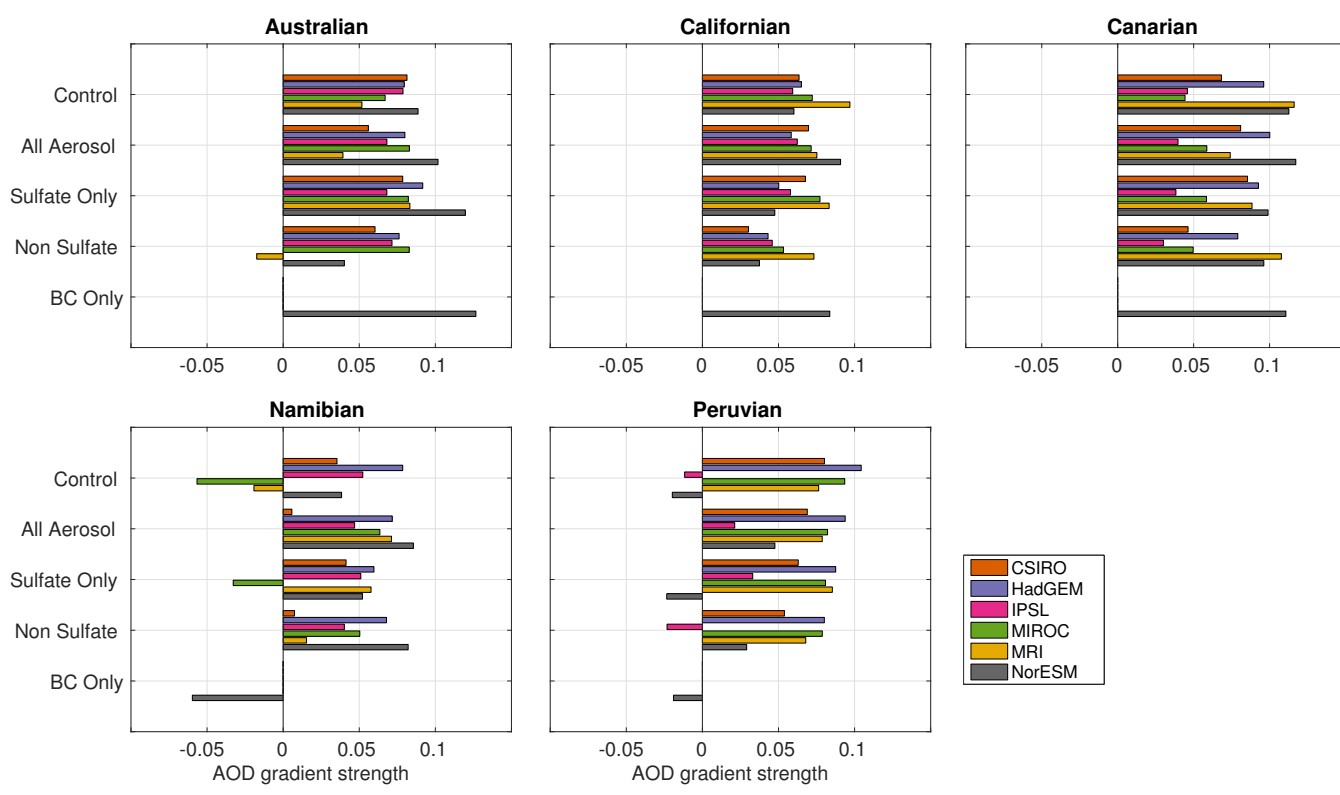

**Figure 5.** AOD gradient strength quantified by the difference of separate linear regressions for the lower and upper 10th percentile of AOD anomaly points respectively. Values are given for all six CMIP5 models for the Australian, Californian, Canarian, Namibian and Peruvian regions and the Control, All Aerosol, Sulfate Only, Non Sulfate and BC Only (for NorESM) cases.

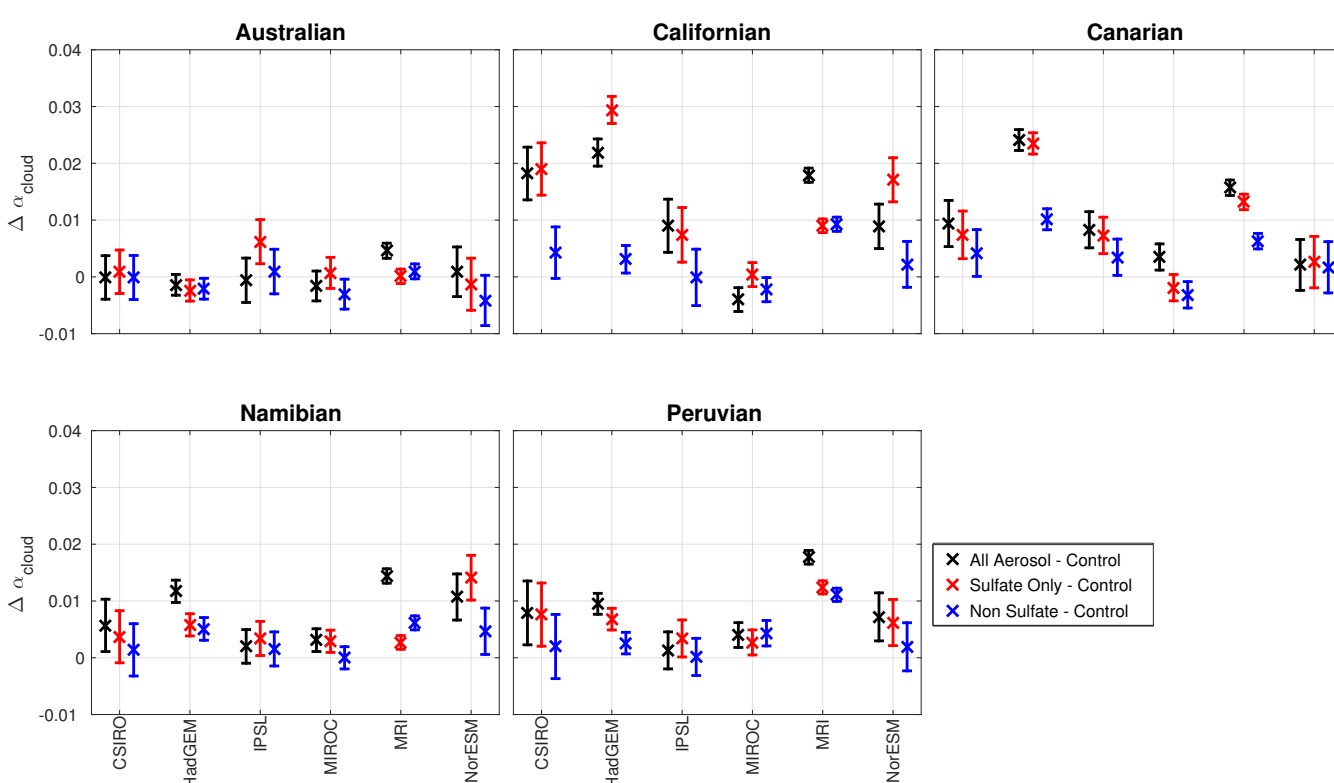

**Figure 6.** Estimated cloud albedo changes in six CMIP5 models for the Australian, Californian, Canarian, Namibian and Peruvian regions, due to changes in emissions of all anthropogenic aerosols (black), sulfate only (red) and non-sulfate (blue). Errorbars indicate one standard deviation.

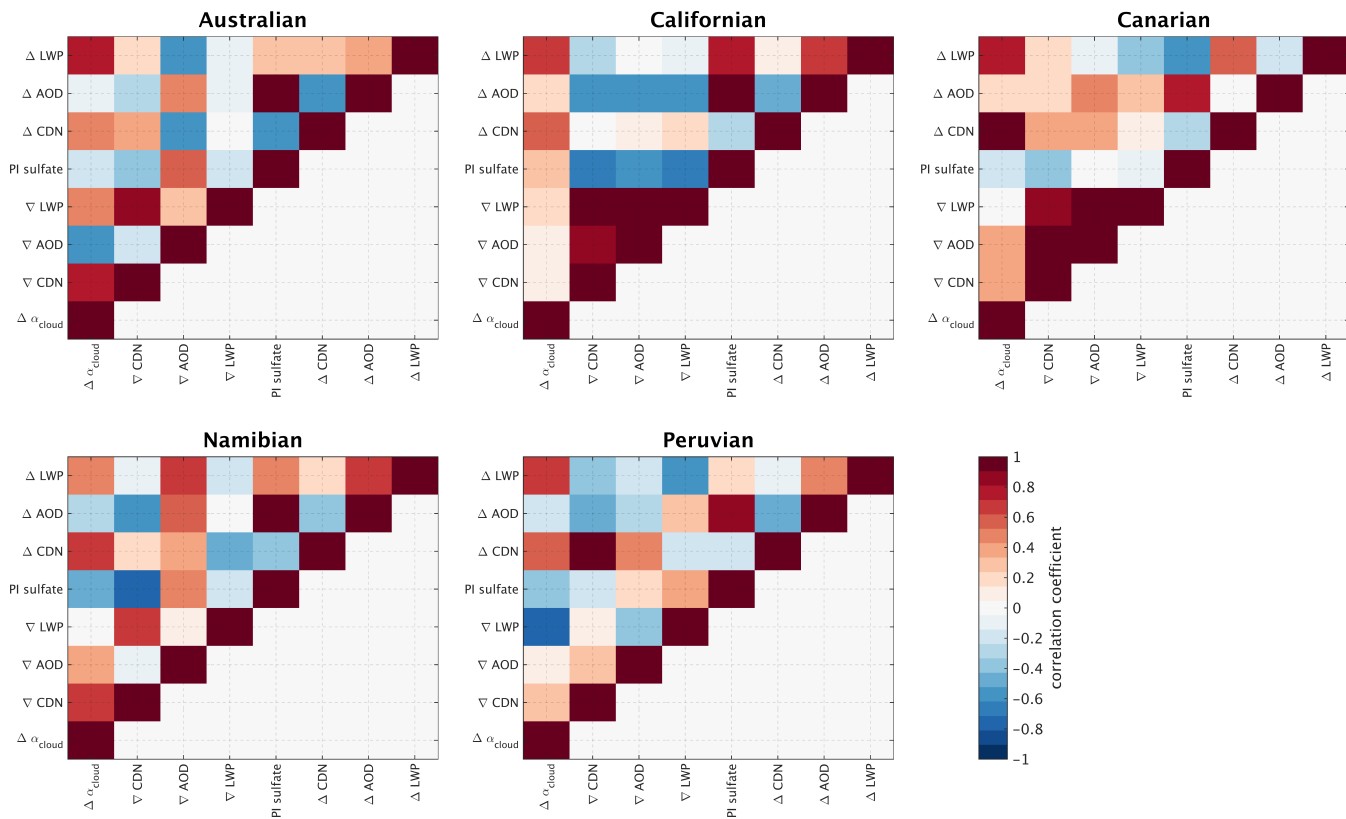

**Figure 7.** Correlation matrices for the Australian, Californian, Canarian and Namibian regions. Regional mean changes in cloud albedo between Control and All Aerosol experiments ($\Delta \alpha_{cloud}$) are related to corresponding changes in CDN, AOD and LWP ($\Delta CDN$, $\Delta AOD$ and $\Delta LWP$ respectively), and to preindustrial sulfate load (PI sulfate), and to gradient strength (month-to-month sensitivity) of CDN, AOD and LWP in the Control case ($\nabla CDN$, $\nabla AOD$ and $\nabla LWP$ respectively), for five CMIP5 models. IPSL does not provide CDN fields, and is excluded from the correlation calculations.

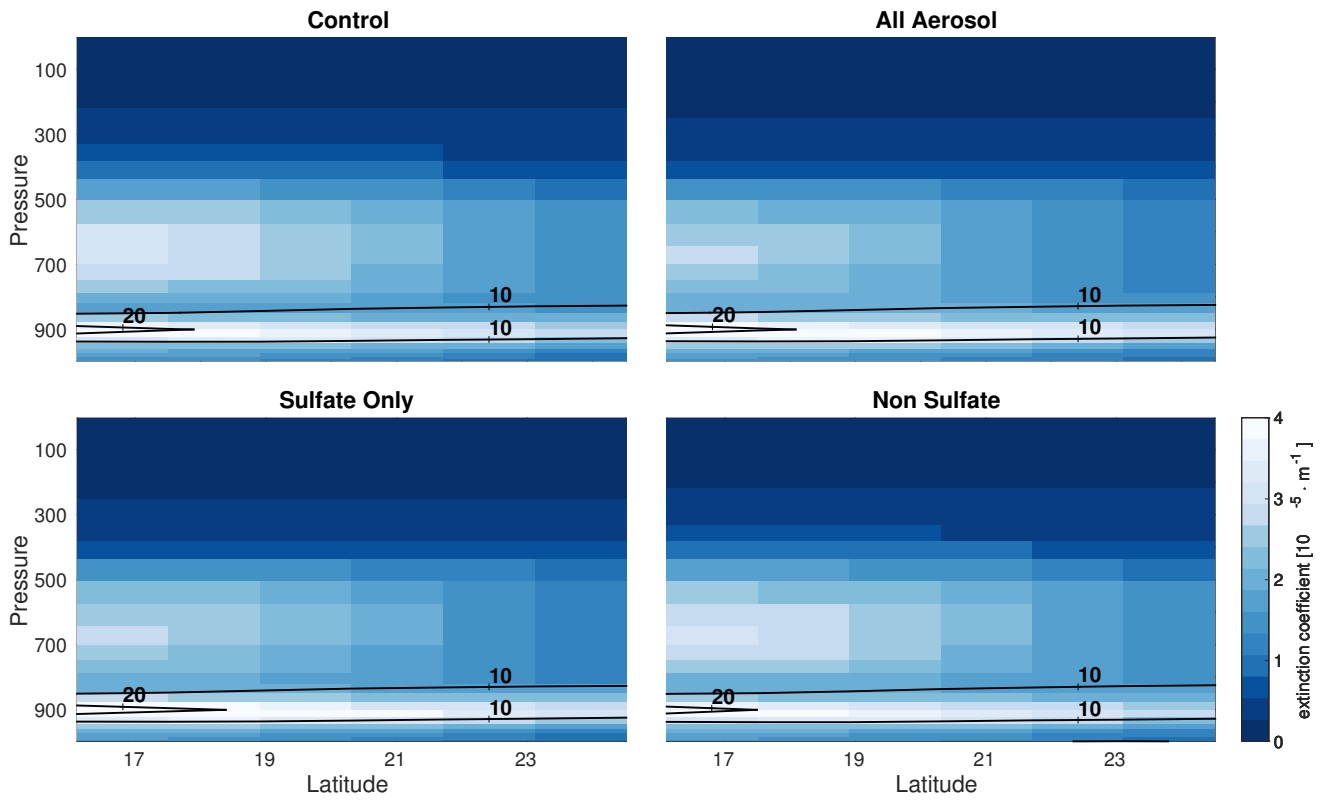

**Figure 8.** Vertical distribution of the aerosol extinction coefficient at 550 nm (colour) and the cloud fraction in % (contours) for the model MIROC in the Canarian region.

**Table 1.** Models considered in the study, where the short names listed are used through.

| Model | Short name | Institute | Resolution | Cloud albedo/ lifetime effect | Reference |
|---|---|---|---|---|---|
| CSIRO-Mk3-6-0 | CSIRO | CSIRO-QCCCE | T63 L18 | yes/ yes | Rotstayn et al. (2012) |
| HadGEM2-A | HadGEM | MOHC | N96 L38 | yes/ yes | Bellouin et al. (2007); Collins et al. (2011) |
| IPSL-CM5A-LR | IPSL | IPSL | 96 x 95 x 39 | yes/ no | Dufresne et al. (2013) |
| MIROC5 | MIROC | MIROC | T85 L40 | yes/ yes | Takemura et al. (2005); Watanabe et al. (2011) |
| MRI-CGCM3 | MRI | MRI | TL159 L48 | yes/ yes | Yukimoto et al. (2012) |
| NorESM1-M | NorESM | NCC | f19 L26 | yes/ yes | Iversen et al. (2013); Kirkevåg et al. (2013) |

**Table 2.** Relative contributions to the total loading in % for the Control experiment. OM for NorESM includes only primary organic aerosol and for HadGEM only secondary organic aerosol.

| Region | Model | Relative loading contributions | | | | |
|---|---|---|---|---|---|---|
| | | sulfate | BC | OM | dust | sea salt |
| Australian | CSIRO | 6 | < 1 | 11 | 28 | 54 |
| | HadGEM | 1 | < 1 | 1 | 34 | 64 |
| | IPSL | 2 | < 1 | 2 | 7 | 89 |
| | MIROC | 4 | < 1 | 3 | 43 | 50 |
| | MRI | 1 | < 1 | 1 | 5 | 93 |
| | NorESM | 5 | < 1 | 16 | 11 | 67 |
| Californian | CSIRO | 6 | < 1 | 10 | 42 | 41 |
| | HadGEM | 1 | < 1 | 2 | 27 | 70 |
| | IPSL | 4 | < 1 | 6 | 29 | 60 |
| | MIROC | 3 | < 1 | 4 | 67 | 26 |
| | MRI | 2 | < 1 | 1 | 10 | 87 |
| | NorESM | 6 | 1 | 20 | 39 | 35 |
| Canarian | CSIRO | 1 | < 1 | 1 | 93 | 5 |
| | HadGEM | 1 | < 1 | 1 | 68 | 30 |
| | IPSL | 1 | < 1 | 1 | 87 | 12 |
| | MIROC | 1 | < 1 | 1 | 92 | 6 |
| | MRI | < 1 | < 1 | < 1 | 78 | 21 |
| | NorESM | 2 | < 1 | 3 | 87 | 8 |
| Namibian | CSIRO | 2 | < 1 | 7 | 78 | 13 |
| | HadGEM | 1 | < 1 | 4 | 41 | 54 |
| | IPSL | 2 | < 1 | 6 | 62 | 29 |
| | MIROC | 3 | < 1 | 8 | 57 | 32 |
| | MRI | 2 | 1 | 5 | 13 | 79 |
| | NorESM | 5 | 1 | 31 | 21 | 41 |
| Peruvian | CSIRO | 4 | < 1 | 7 | 66 | 23 |
| | HadGEM | 2 | < 1 | 4 | 20 | 74 |
| | IPSL | 5 | < 1 | 7 | 8 | 80 |
| | MIROC | 8 | < 1 | 7 | 27 | 59 |
| | MRI | 3 | < 1 | 2 | 2 | 93 |
| | NorESM | 10 | < 1 | 31 | 10 | 49 |