# Peer review of "Cloud albedo changes in response to anthropogenic sulfate and non-sulfate aerosol forcings in CMIP5 models"

_Atmospheric Chemistry and Physics, 2016_

## Referee Comment (RC1) · Anonymous Referee #1 · 8 Feb 2017

The authors use the CMIP5 sstClim, sstClimAerosol, and sstClimSulfate experiments to analyze the effect of anthropogenic aerosol emissions on cloud albedo in the marine stratocumulus regions.

The first major part of the paper is the temporal AOD variability on monthly scales in the CMIP5 models in these regions. The results (that the anthropogenic AOD perturbation is small compared to the temporal variability) are presented well and, in my opinion, should be published.

The second major part of the paper uses the cloud-fraction–scene-albedo technique to analyze aerosol perturbations to the cloud albedo. I agree with the authors that this formalism is well suited to the study of aerosol–cloud interactions. I also think that

applying the formalism to these CMIP5 experiments is a worthwhile way to investigate the effect of different aerosol species in the participating models.

The major shortcoming of the study is the use of AOD as the aerosol variable. The authors cite Andreae et al. (2009) to justify this choice, but convincing evidence since then shows that AOD is a poor proxy for the aerosol properties that matter for aerosol–cloud interactions (CCN concentrations). As a consequence, the results are inconclusive and the discussion section becomes difficult to follow.

My suggestion to the authors is to avail themselves of the advantage that a modeling study affords them – that the CCN and CDNC fields are provided. Investigating the albedo response to the anthropogenic perturbations in these variables should provide much more easily interpretable results, including the results that the authors are after (relative influence of different species, monthly variability compared to the anthropogenic perturbation). The results based on AOD could still be retained; they would show what differences are to be expected when the CCN field is known versus when only the AOD is known.

In light of this fairly major suggested revision, I am not providing detailed comments now, but I would be happy to do so on the revised version.

---

## Referee Comment (RC2) · Anonymous Referee #2 · 20 Mar 2017

In this paper, AMIP experiments with changes in all aerosol, sulphate aerosol, and non-sulfate aerosol are used to determine the effects of different aerosol types on the albedo of marine stratocumulus clouds. The main conclusion seems to be that the present day vs. pre-industrial changes in AOD due to anthropogenic aerosol, and hence any changes they induce in in cloud albedo, are small compared to AOD variability. The authors also find that, in their set of models, changes in cloud water content are more important to changes in cloud albedo than changes in AOD.

I found that the methodology was not always well explained, and that the approach was not always well justified. The authors discuss the parameterisation of cloud albedo effect in their models, but then make a strange choice to use AOD as their aerosol

metric, despite the fact that this does not directly relate to cloud albedo in the models they use. This makes interpretation of their results very difficult. Aspects of the analysis presented in the paper suggests to me that the authors have the data to revise the study so that their analysis approach is better suited to the questions they are trying to answer, and I will discuss this in more detail below. As such, this paper may be suitable for publication following major revisions.

While the manuscript is reasonably written, it lacks a clear narrative. It will benefit from a clearer statement of the objectives, methodology, and conclusions. However, most of all, it will benefit from an approach that better relates aerosol changes to cloud albedo. I suspect that the issues with the writing will resolve themselves with improvements to the analysis, but the authors should bear this in mind when revising the paper.

Major comments This study adopts the method of Bender et al. (2016) for relating AOD to cloud albedo. However, there is no summary of this approach provided here, so that I had to refer to Bender et al. (2016) to know what the method was. I would like to see a revised version of the manuscript that is more self-contained in this respect.

The authors have clearly taken the time to look at the parameterisation of the cloud albedo effect in their set of models, commenting on it on page 4, line 8, and again later in the paper. They correctly state that the parameterisation schemes differ in complexity, but don't comment on the fact that they also relate different aerosol species to cloud albedo. Most of my major issues with the manuscript relate to the authors approach to these parameterisation schemes. Firstly, the schemes in all the models used relate cloud droplet number concentration, not AOD, to cloud albedo. I am not convinced that AOD is trivially related to cloud droplet number concentration, and the authors make no real attempt to demonstrate this. Secondly, and most importantly, not all aerosol species affect cloud albedo in the models. Crucially for this work, dust does not directly interact with cloud albedo in any of the models considered here, despite accounting for most of the AOD, and its variability. A third point relates to this: cloud droplet number concentrations are related to different species in the different models. For example,

HadGEM2-ES uses only sulphate and sea salt, while CSIRO-Mk3-6-0 also includes carbonaceous aerosol, and sea salt does not contribute to cloud droplet number concentration in IPSL (e.g. Wilcox et al., 2015; references in Table 1 of the submitted manuscript). These differences will have big influence on the regional responses in these models.

I think it is important that the authors repeat their analysis with an aerosol metric that is more closely related to the cloud albedo in the models: cloud droplet number concentration, or species-specific vertically integrated load. The authors already present some analysis of sulphate load, so I hope that this is not too onerous for them, and that it improves their results as I expect it will.

Minor comments Page 2, Line 5: I would need to read Ackerman et al., but my understanding is that absorbing aerosols reduce cloud cover by causing a local heating, rather than anything to do with their efficiency as CCN. Since the role of absorbing aerosol is key to the discussion later in the paper, it would be nice for the authors to clarify this a bit more.

Page 3, line 2 (and later): The authors state that absorbing aerosols overlying the cloud is not well represented in models. This is a point that is revisited later. As this is one of the things investigated in the paper, I would like to see a bit more background on this (not necessarily at this point in the paper). Is this poorly represented because of shortcomings in the modelled cloud distribution, aerosol distribution, aerosol properties, etc.? How might these things affect the result? My suspicion is that it is a combination of several factors, including a tendency of models to underestimate the amount of aerosol above the cloud layer (e.g. Peers et al., 2016), and to underestimate atmospheric heating due to carbonaceous aerosols (e.g. Myhre and Samset, 2015).

Page 3, line 13 (and later): The authors mention the large inter-model variations in global-mean AOD. For the interpretation of their results, I think it is important to highlight

that properties such as aerosol mass and number, which are more closely related to modelled cloud albedo, are even more diverse, and not so easily tuned to observations.

Page 3, line 30: Aerosol particles are not parameterised in models, but their interactions with clouds are. However, modelled aerosols are idealised compared to the real world, so perhaps that is what the authors mean here.

Page 5, line 26: Dust and sea salt account for most of the mass, but do they account for most of the change?

Page 6, lines 16 to 26: This section could benefit from a bit more rigour. Are differences significant?

Page 7: The section is lacking mentions of dust, which you might expect to be important in these regions. In the models, it won't affect the cloud albedo, but it might affect the clouds by changing the heating profiles.

Page 8, line 16: How does this change compare to the preindustrial values?

Page 9, line 15: The authors suggest that LWP is more important for cloud albedo than anthropogenic aerosol change, and should include some context from previous publications here, e.g. Gettleman (2015).

Page 11, line 23: Species dependence is in fact hard-coded in the models.

Figure 6/7: One of the main conclusions in the text is that the correlations in Figure 7 are much stronger than those in Figure 6. The authors should quantify this, as it is not obvious by eye.

Table 3: The caption says that all of the correlation coefficients in this table are significant at the 95% level (this should say 5% level). I find this surprising given the magnitude of some of the coefficients (0.01, 0.02, 0.03,...), and what I think is only a small number of data points (20 years for sstClim experiments? – this should be stated in the text).

Gettelman (2015) Putting the clouds back in aerosol–cloud interactions doi: 10.5194/acp-15-12397-2015

Myhre and Samset (2015) Standard climate models radiation codes underestimate black carbon radiative forcing doi: 10.5194/acp-15-2883-2015

Peers et al. (2016) Comparison of aerosol optical properties above clouds between POLDER and AeroCom models over the South East Atlantic Ocean during the fire season, doi: 10.1002/2016GL068222

Wilcox et al. (2015) Quantifying sources of inter-model diversity in the cloud albedo effect doi: 10.1002/2015GL063301

---

## Author Comment (AC1) · 1 May 2017

We thank reviewer 1 for his/her comments. We uploaded the response as a supplement, including a difference-pdf.

Please also note the supplement to this comment:
http://www.atmos-chem-phys-discuss.net/acp-2016-1152/acp-2016-1152-AC1-supplement.zip

---

## Author Response (AR1)

The authors use the CMIP5 sstClim, sstClimAerosol, and sstClimSulfate experiments
to analyze the effect of anthropogenic aerosol emissions on cloud albedo in the marine
stratocumulus regions.
The first major part of the paper is the temporal AOD variability on monthly scales in the
CMIP5 models in these regions. The results (that the anthropogenic AOD perturbation
is small compared to the temporal variability) are presented well and, in my opinion,
should be published.

The second major part of the paper uses the cloud-fraction–scene-albedo technique
to analyze aerosol perturbations to the cloud albedo. I agree with the authors that this
formalism is well suited to the study of aerosol–cloud interactions. I also think that
applying the formalism to these CMIP5 experiments is a worthwhile way to investigate
the effect of different aerosol species in the participating models.
The major shortcoming of the study is the use of AOD as the aerosol variable. The authors
cite Andreae et al. (2009) to justify this choice, but convincing evidence since then
shows that AOD is a poor proxy for the aerosol properties that matter for aerosol–cloud
interactions (CCN concentrations). As a consequence, the results are inconclusive and
the discussion section becomes difficult to follow.
My suggestion to the authors is to avail themselves of the advantage that a modeling
study affords them – that the CCN and CDNC fields are provided. Investigating
the albedo response to the anthropogenic perturbations in these variables should provide
much more easily interpretable results, including the results that the authors are
after (relative influence of different species, monthly variability compared to the anthropogenic
perturbation). The results based on AOD could still be retained; they would
show what differences are to be expected when the CCN field is known versus when
only the AOD is known.
In light of this fairly major suggested revision, I am not providing detailed comments
now, but I would be happy to do so on the revised version.

We would like to thank the reviewer for his/her comments. We addressed all the major revision suggestions and added additional
analysis for the cloud droplet number fields. The reviewer is right, that AOD is not the best indicator of aerosol properties that matter
to aerosol-cloud interactions.  CDNC (vertically resolved, or at cloud top) is not available for most models, but all models but one
provide the column integrated cloud droplet number (unit 1/m2), and we have now investigated these fields closer.

Our motivation for using AOD was not clear in the manuscript, but has now been made more explicit. AOD is indeed an integral
measure, including "CCN-active" aerosol, as well as other aerosol. Thereby, the AOD-relation to cloud albedo and scene albedo may
be different in different regions, dominated by different aerosol types. This is also what previous work has shown to be the case when
looking at observations, while models have seemed to have a less refined view of AOD overall being positively related to cloud
albedo (Bender et al. 2016). Separating aerosol species in models as we do here is a way to search for some refinement in the
models – is AOD still always positively related to cloud albedo, or can AOD be weakly or negatively related to cloud albedo,
depending on dominating aerosol type?

CDNC in an albedo-cloud fraction diagram should always yield a positive gradient, and this is indeed the case, as indicated by the
new Figure 3 in the revised manuscript.
That is to say that for a given cloud fraction, a cloud with more droplets has a higher reflectivity. The same can be said of LWP, which
was also shown in Bender et al. (2016), for satellite observations and present-day model simulations (their Figures 6-7, respectively).
The sstClim simulations studied here show the same LWP-pattern, as we now mention in the text.
The AOD-gradient on the other hand may vary, as the scene albedo for a given cloud fraction is dependent both on how the present
aerosols act as CCN, how they affect the clear-sky albedo, and how different aerosol types are distributed in the vertical. If, for
instance, models underestimate absorbing aerosol presence and/or reflectivity above clouds, then they may yield a positive relation
between AOD and cloud albedo (a positive gradient in albedo- cloud fraction space), when a more realistic representation of
absorbing aerosols should actually give a weak or reversed gradient.

We have made changes primarily to the Introduction of the paper to explain this. We have also taken care to better explain the albedo-cloud fraction technique used, rather than just referencing previous studies. We have also added analysis of CDN in relation to albedo and cloud fraction, as well as a more complete analysis of relations between AOD, CDN, LWP, sulfate loading, and cloud albedo (see new Figure 7), highlighting further model differences, but indicating which parameter changes have the greatest effect on cloud albedo changes between simulations.

Note: A new output version for the model CSIRO was provided on the ESGF data archive and was used for the analysis in the revised manuscript.

**Anonymous Referee #2**

In this paper, AMIP experiments with changes in all aerosol, sulphate aerosol, and non-sulfate aerosol are used to determine the effects of different aerosol types on the albedo of marine stratocumulus clouds. The main conclusion seems to be that the present day vs. pre-industrial changes in AOD due to anthropogenic aerosol, and hence any changes they induce in in cloud albedo, are small compared to AOD variability. The authors also find that, in their set of models, changes in cloud water content are more important to changes in cloud albedo than changes in AOD.

I found that the methodology was not always well explained, and that the approach was not always well justified. The authors discuss the parameterisation of cloud albedo effect in their models, but then make a strange choice to use AOD as their aerosol metric, despite the fact that this does not directly relate to cloud albedo in the models they use. This makes interpretation of their results very difficult. Aspects of the analysis presented in the paper suggests to me that the authors have the data to revise the study so that their analysis approach is better suited to the questions they are trying to answer, and I will discuss this in more detail below. As such, this paper may be suitable for publication following major revisions.

While the manuscript is reasonably written, it lacks a clear narrative. It will benefit from a clearer statement of the objectives, methodology, and conclusions. However, most of all, it will benefit from an approach that better relates aerosol changes to cloud albedo. I suspect that the issues with the writing will resolve themselves with improvements to the analysis, but the authors should bear this in mind when revising the paper.

We would like to thank the reviewer for his/her comments. We have revised the manuscript in Accordance with the reviewer's suggestions, and address each of the comments in the following.

Major comments This study adopts the method of Bender et al. (2016) for relating AOD to cloud albedo. However, there is no summary of this approach provided here, so that I had to refer to Bender et al. (2016) to know what the method was. I would like to see a revised version of the manuscript that is more self-contained in this respect.

We have made changes in the Introduction to explain the methodology clearer.

The authors have clearly taken the time to look at the parameterisation of the cloud albedo effect in their set of models, commenting on it on page 4, line 8, and again later in the paper. They correctly state that the parameterisation schemes differ in complexity, but don't comment on the fact that they also relate different aerosol species to cloud albedo. Most of my major issues with the manuscript relate to the authors approach to these parameterisation schemes. Firstly, the schemes in all the models used relate cloud droplet number concentration, not AOD, to cloud albedo. I am not convinced that AOD is trivially related to cloud droplet number concentration, and the authors make no real attempt to demonstrate this. Secondly, and most importantly, not all aerosol species affect cloud albedo in the models. Crucially for this work, dust does not directly interact with cloud albedo in any of the models considered here, despite accounting for most of the AOD, and its variability. A third point relates to this: cloud droplet number concentrations are related to different species in the different models. For example, HadGEM2-ES uses only sulphate and sea salt, while CSIRO-Mk3-6-0 also includes carbonaceous aerosol, and sea salt does not contribute to cloud droplet number concentration in IPSL (e.g. Wilcox et al., 2015; references in Table 1 of the submitted manuscript). These differences will have big influence on the regional responses in these models.

I think it is important that the authors repeat their analysis with an aerosol metric that is more closely related to the cloud albedo in the models: cloud droplet number concentration, or species-specific vertically integrated load. The authors already present some analysis of sulphate load, so I hope that this is not too onerous for them, and that it improves their results as I expect it will.

We would like to thank the reviewer for his/her comments. We realize that the way the manuscript was written, our intent with using AOD was not clear, and we have made changes to improve this, and included analysis of cloud droplet number, as suggested. As CDNC at cloud top or vertically resolved is not available for most models, we have used the column integrated cloud droplet number (CDN) in unit $1/m^2$, that all models except one provide.

As the reviewer points out, AOD is not trivially related to CDNC and not all aerosols that contribute to AOD contribute to CDNC. Accordingly, cloud albedo and scene albedo do not necessarily increase with increasing AOD. The AOD may be dominated by aerosols that are not active as CCN, and may also have contributions from absorbing aerosols that actually decrease the scene albedo if they are above the clouds. This is what satellite observations have indicated in previous studies, whereas models seem to have more trouble making these distinctions (Bender et al. 2016). In this study we separate aerosol types to look closer at these relations in the models.

We have made changes to the text according to the points made by the reviewer making clarifications regarding aerosol species contributing to AOD vs. contributing to CDNC, and the relation between AOD and CDNC. (see Section 2, Section 3.2 )

We have also added analysis of CDN, parallel to the AOD-analysis, see Figures 3 and 7. As expected, the CDN consistently shows a positive gradient in albedo-cloud faction space (see new Figure 3).

As the issues raised by the reviewer here are quite similar to the major comment made by Reviewer 1, we also refer to the reply to that comment, for further explanation.

Minor comments Page 2, Line 5: I would need to read Ackerman et al., but my understanding is that absorbing aerosols reduce cloud cover by causing a local heating, rather than anything to do with their efficiency as CCN. Since the role of absorbing aerosol is key to the discussion later in the paper, it would be nice for the authors to clarify this a bit more.
Thank you, this has been clarified.

Page 3, line 2 (and later): The authors state that absorbing aerosols overlying the cloud is not well represented in models. This is a point that is revisited later. As this is one of the things investigated in the paper, I would like to see a bit more background on this (not necessarily at this point in the paper). Is this poorly represented because of shortcomings in the modelled cloud distribution, aerosol distribution, aerosol properties, etc.? How might these things affect the result? My suspicion is that it is a combination of several factors, including a tendency of models to underestimate the amount of aerosol above the cloud layer (e.g. Peers et al., 2016), and to underestimate atmospheric heating due to carbonaceous aerosols (e.g. Myhre and Samset, 2015).
We have expanded this discussion and added the suggested references. We are currently working on a closer investigation of the reasons for discrepancies, and test the sensitivity of the vertical aerosol profile to various parameters in NorESM, similar to the way Kipling et al. 2016 investigate HadGEM3. This reference is also added.

Page 3, line 13 (and later): The authors mention the large inter-model variations in global-mean AOD. For the interpretation of their results, I think it is important to highlight that properties such as aerosol mass and number, which are more closely related to modelled cloud albedo, are even more diverse, and not so easily tuned to observations.
Model diversity in AOD change due to differences in aerosol mass and number and the parameterization of radiative properties of different aerosol types. Properties like aerosol mass and number as well as cloud droplet number are indeed poorly constrained by observations, allowing for large inter-model variation. We have highlighted this in Section 3.1 and 3.2.

Page 3, line 30: Aerosol particles are not parameterised in models, but their interactions with clouds are. However, modelled aerosols are idealised compared to the real world, so perhaps that is what the authors mean here.
Of course, we have replaced the word "parameterised" with "represented".

Page 5, line 26: Dust and sea salt account for most of the mass, but do they account for most of the change?
No, the aerosol types that contribute to the changes between experiments are those with anthropogenic emission sources, sulfate in the Sulfate case, and BC and OM in the Non Sulfate case, as mentioned in Section 2. This has been clarified.

Page 6, lines 16 to 26: This section could benefit from a bit more rigour. Are differences
significant?

We have updated significant statement in the text. All changes are significant except for two models in two different regions.

Page 7: The section is lacking mentions of dust, which you might expect to be important
in these regions. In the models, it won't affect the cloud albedo, but it might affect the
clouds by changing the heating profiles.

Dust is important in some of the regions studied, as mentioned in Sections 3.2, 3.3. and 3.4. In Section 3.4, we now point out that the dust may affect clouds by changing the heating profiles.

Page 8, line 16: How does this change compare to the preindustrial values?

Relative differences between sensitivity experiments and preindustrial case have been updated in the text.

Page 9, line 15: The authors suggest that LWP is more important for cloud albedo
than anthropogenic aerosol change, and should include some context from previous
publications here, e.g. Gettleman (2015).

Thanks for the reference. We added it to Sections 3.3 and 4.

Page 11, line 23: Species dependence is in fact hard-coded in the models.

This has been clarified.

Figure 6/7: One of the main conclusions in the text is that the correlations in Figure 7
are much stronger than those in Figure 6. The authors should quantify this, as it is not
obvious by eye.

Figures 6-7 have been replaced with a new figure, showing a correlation matrix between different variables and investigating further their relation to cloud albedo changes

Table 3: The caption says that all of the correlation coefficients in this table are significant
at the 95% level (this should say 5% level). I find this surprising given the
magnitude of some of the coefficients (0.01, 0.02, 0.03,. . .), and what I think is only a
small number of data points (20 years for sstClim experiments? – this should be stated
in the text).

This table has been removed. The new figure 7 includes the correlation between AOD and pre-industrial sulfate loading. The focus is on sulfate loading rather than other aeorosl types, since it is expected to be the main driver of AOD changes between the used CMIP5 experiments.

Gettelman (2015) Putting the clouds back in aerosol–cloud interactions doi:
10.5194/acp-15-12397-2015
Myhre and Samset (2015) Standard climate models radiation codes underestimate
black carbon radiative forcing doi: 10.5194/acp-15-2883-2015
Peers et al. (2016) Comparison of aerosol optical properties above clouds between
POLDER and AeroCom models over the South East Atlantic Ocean during the fire
season, doi: 10.1002/2016GL068222
Wilcox et al. (2015) Quantifying sources of inter-model diversity in the cloud albedo
effect doi: 10.1002/2015GL063301

Thank you for the references. We added the suggested new references.

Note: A new output version for the model CSIRO was provided on the ESGF data archive and was used for the analysis in the revised manuscript.

[revised manuscript text omitted]

---

## Author Response (AR2)

**Submitted on 30 May 2017**

The main parts of the revised paper, in my understanding, are:

1. Analysis of the temporal variability of preindustrial and present-day AOD and cloud droplet number on monthly scales in CMIP5 models in stra- tocumulus regions; natural variability is found to be large compared to the anthropogenic perturbation, and the anthropogenic change in CDN is found to depend most strongly on anthropogenic sulfate AOD.

2. Use of the cloud-fraction–albedo framework to investigate the effect of AOD and vertically integrated cloud droplet number on cloud albedo in CMIP5 models; the relationship is stronger than in satellite observations, but the authors do not come to any systematic conclusions about the representation of aerosol–cloud interactions in the models investigated in this study. Both in content and in clarity of the presentation, this version is a major improvement over the first submission. An application of the cloud-fraction– albedo framework to the CMIP5 models has not been published before, and the authors' findings in both main parts of the paper are worthy of publication. There are, however, a few points that should still be addressed. Methodological comments (page and line numbers refer to the track-changes manuscript supplied as part of the author response):

We would like to thank referee #1 for his/her comments. We respond to each of the questions and comments in the following (comment in black font, reply in blue).

- I like the addition of cloud droplet number to the analysis. However, I expect the vertically integrated cloud droplet number to be strongly correlated with cloud thickness (which in turn strongly affects the cloud albedo). Thus, "cloud brightening" due to CDN is probably a mixture of CDNC and LWP brightening, but the paper treats it as purely CDNC brightening. For most of the models used in this study (except for CSIRO and NorESM), the CMIP5 archive also contains the cloud-top CDNC (variable name cldncl); I think the authors need to test whether their conclusions based on CDN also apply when they use CDNC, or otherwise reword the manuscript significantly.

  Thank you for the comment. The CMIP5 variable 'cldncl' is available only for the three models MIROC, MRI and CSIRO, not for IPSL, HadGEM and NorESM (see the different ESGF nodes for CMIP5 output). Since more models provided the column integrated cloud droplet number (CDN) and not CDNC at cloud-top, we decided to present the results mainly for CDN fields. But we added some discussion of the CDNC at cloud-top fields (see Page 8, L14 ff. and Page 13, L7 ff.).

- Abstract, last sentence: in the usual taxonomy, aerosol–cloud interactions are divided into a CDNC effect, an LWP effect, and a cloud fraction effect. Since the cloud albedo is evaluated at fixed f c = 1, I don't see what other variables apart from LWP and CDNC the albedo could depend on.

  This is true, and the sentence was unclear. The analysis of the AOD anomaly in the cloud fraction- albedo space was conducted to see whether

the month-to-month variability of AOD (through its link to aerosol number and CCN and CDNC) in the models is related to their strength in cloud albedo changes between the experiments and found that this was not of major importance. We changed the sentence in the abstract.

- In Figs. 3 and 4, the three fit lines always appear to have the same y intercept. Are the fits constrained to the same y intercept? If so, does it make a difference to the results whether this constraint is enforced? If so, is there a conceptual reason to prefer one over the other?

  Yes, the fits within each experiment are forced to the same y-intercept. Since we are interested in the relative position of "high" and "low" values (red and blue lines) this does not make a difference to our results. The fit is not forced to a common y-intercept between experiments. We applied the same method to quantify the gradient strength for all experiments, so that we do not expect a great difference from using a non-forced intercept. The variability of clear-sky albedo within one experiment is small with standard deviations less than 0.01.

- Conclusions, p. 14, l. 32: since the gradients are not normalized to the AOD variability, the dust-dominated regions can have large gradients even if the model representation of ACI is insensitive to dust. This paragraph might present a stronger conclusion if you looked at the gradient normalized to the spread in AOD.

  Thank you for your suggestion. Figure R1 shows the normalized gradient strength (normalized by one standard deviation). The correlation between cloud albedo changes and different factors including normalized AOD, LWP and CDN gradients are presented in Figure R2.

  In the gradient-plots (Figs. 3 and 4 in the manuscript), we account for variability within each region by using de-seasnoalized and de-regionalized data for the AOD anomaly gradient. The variability plots (Fig. 2 in the manuscript ) show the total variability, where seasonality contributes significantly e.g. in the Canarian region. The gradient plots also show AOD anomaly for a given cloud fraction. For these regions, we find it more straight-forward to show the gradient strengths without further normalization and found that the aerosol composition is more important for the gradient strength than the AOD variability (see discussion on Page 8, L32 in the revised manuscript).

- It would be interesting to see the equivalent of Fig. 3 or 4 for a region or model where the AOD or CDN gradient is not consistently positive.

  You can see in Figure 3 the AOD gradient for the model MIROC in the Namibian region as an example. We decided to not include a figure with a negative gradient in the manuscript, since Figure 5 gives an overview of all gradients. The CDN gradients were found to be positive for all models in all regions and experiments (see Page 8, L12 in the revised manuscript)

Comments on the presentation (page and line numbers refer to the track-changes manuscript supplied as part of the author response):

- p. 3, l. 12: the phrase "second order" is interesting enough that I would like to see a few words of definition

Second-order variability refers here to the variability in albedo at a given cloud fraction, i.e. variations in albedo driven by other factors, inlcuding cloud microphysics. We rephrased the sentence.

- l. 21 ff.: Is it possible to link this result to the underestimate of aerosol ERF from present-day variability.

  As we understand this comment, then yes, it might be true that a stronger relation between AOD and CDN in models than in observations might be related to an overestimate in ERF in models.

- p. 9, l. 21: "all regions and all experiments" – from the caption, I thought Fig. 3 only showed one region

  Yes, the sentence was misleading. Figure 3 in the manuscript shows the CDNC gradient only for the model NorESM in the Californian region. We found a positive CDN gradient for all models and experiments in all regions. We rephrased the sentence.

- p. 10, l. 21: where does the number 8% come from?

  We quantified cloud albedo changes by calculating relative changes of cloud albedo between the sensitivity experiments and the control experiment. The highest change with 8% was found for the model HadGEM for a relative change between the 'Sulfate Only' and Control experiment. We changed the sentence.

- Fig. 2: A logarithmic vertical scale would improve the legibility of the non-dusty panels and make it easier to follow the discussion in Sec. 3.2 visually.

  Thank you for the suggestion. To compare the different models, the logarithmic scale would indeed improve the legibility. But since our analysis focuses on regional differences as well, a non-logarithmic plot facilitates the regional inter-comparison.

- Figs. 3 and 4: these figures are swapped; also, if it's not too much trouble, it would be useful for the reader to see a present-day satellite plot for comparison.

  Figures 3 and 4 have been swapped. Figure 3 in the manuscript shows now the correct plot of CDN anomaly in the albedo-cloud fraction space and Figure 4 the AOD anomaly. The AOD gradient plot for present-day satellite data is attached in Figure 4. We decided to not include this figure in the manuscript, since we reference Bender et al. (2016), where satellite data is presented. Our results are consistent with Bender et al. (2016) with reversed gradients in the Canarian and Namibian region and no organized pattern in the Australian, Californian and Peruvian region.

- Is there a reason for switching the horizontal and vertical axes between Figs. 2 and 5?

  There is no particular reason, but we found it more intuitive to represent the gradient strength as horizontal bars and found it easier to compare experiments by presenting them from top to bottom.

- I am not a huge fan of the use of $\nabla$ as an abbreviation for gradient (it should be reserved for actual spatial gradients), and having $\nabla$ and $\Delta$ appear on the same plot is a bit of a cognitive challenge, but I understand that some shorthand is necessary. It may help to reorder the variables so that the $\nabla$ s all appear before the $\Delta$ s.

  We changed the order in Figure 7 in the manuscript.

- The newly added text in particular could use a thorough proofreading; the bibliography has the standard problems (inconsistent capitalization, inconsis- tent abbreviation of journal names).

  We corrected the bibliography.

Note: Figure 2 has been updated to correct for a slight shift in the geographical areas for two of the regions.

[Figure]

Figure R1: Normalized AOD gradient strength quantified by the difference of separate linear regressions for the lower and upper 10th percentile of AOD anomaly points respectively. Values are given for all six CMIP5 models for the Australian, Californian, Canarian, Namibian and Peruvian regions and the Control, All Aerosol, Sulfate Only, Non Sulfate and BC Only (for NorESM) cases.

[Figure]

Figure R2: Correlation matrices for the Australian, Californian, Canarian and Namibian regions. Regional mean changes in cloud albedo between Control and All Aerosol experiments ($\Delta\alpha_{cloud}$) are related to corresponding changes in CDN, AOD and LWP ($\Delta CDN$, $\Delta AOD$ and $\Delta LWP$ respectively), and to preindustrial sulfate load (PI sulfate), and to normalized gradient strength (month-to-month sensitivity) of CDN, AOD and LWP in the Control case ($\nabla CDN$, $\nabla AOD$ and $\nabla LWP$ respectively), for five CMIP5 models. IPSL does not provide CDN fields, and is excluded from the correlation calculations.

[Figure]

Figure R3: AOD anomaly gradient for the model MIROC for all experiments, in the Namibian region. Two linear regressions are performed for the separated upper (red dashed line) and lower (blue dashed line) 10% of the data. The black dashed line represents a linear fit for all the AOD anomaly data, and the estimated cloud albedo is derived from the slope and intercept of that linear regression. The color scale for the anomaly was normalized by the standard deviation of AOD for each cloud fraction bin.

[Figure]

Figure R4: AOD anomaly gradient for MODIS and CERES satellite data for the Australian, Californian, Canarian, Namibian and Peruvian regions. Two linear regressions are performed for the separated upper (red dashed line) and lower (blue dashed line) 10% of the data. The black dashed line represents a linear fit for all the AOD anomaly data, and the estimated cloud albedo is derived from the slope and intercept of that linear regression. The color scale for the anomaly was normalized by the standard deviation of AOD for each cloud fraction bin.

**Anonymous Referee # 2**

**Submitted on 4 June 2017**

Second review of Frey et al., 'Cloud albedo changes in response to anthropogenic sulphate and non-sulfate aerosol forcings in CMIP5 models.
The authors have made improvements to the manuscript since the last review. In particular, the methodology is clearer, and they have moved away from using only AOD as an aerosol metric. The article should be published, following a few more minor revisions.
We would like to thank referee #2 for his/her comments. We respond to each of the questions and comments in the following (comment in black font, reply in blue).

- Page 1, line 10: 'The magnitude of cloud albedo changes in response to the aerosol changes are most closely related to changes in cloud droplet number and liquid water path in the models.' This seems obvious to me – my understanding is that these are the only related variables in models. In most climate models reff is a function of CDNC and LWP, optical depth is a function of reff and LWP, and albedo is a function of optical depth. It would be much more interesting to know if one of reff or LWP is more closely related to the change in albedo.

  Thank you for your comment. We changed the sentence. The sentence was indeed not drawing attention to our conclusions, that we found the month-to-month AOD variability in our set of models and experiments to have no clear relation to the strength of cloud albedo changes between experiments. We added some discussion about CDNC at cloud-top and effective radius to analyze if cloud albedo changes are due to cloud thickening or caused only by effective radius modifications (see Page 8, L13 ff. and Page 13, L8 ff.). The CDNC at cloud-top fields were only provided for a subset of our models, which makes it difficult to draw robust conclusions.

- Page 4, L30: This repeats the sentence at line 27, but is phrased much better. I recommend moving this to line 27, in place of the original L27 sentence.

  The sentence has been moved according to the suggestion.

- Page 5, L19: I suspect there is a degree of model tuning behind this statement. Model AODs are much more uniform than the aerosol loadings, where even sulfate can have a factor of 4 spread in the global mean (e.g. Wilcox et al., 2015). Since you focus on the indirect effect, I think it is worth highlighting this difference, as you are likely to see effects of it in your results. Of course, AOD is easier to observe than aerosol loads, so it is harder to know which model is closer to the truth in this case, but an awareness of model diversity in aerosol load is useful for interpretation of the modeled cloud changes.

  Thank you for the comment. We are not aware of AOD being a main explicit tuning parameter in the models we are using. We added a sentence addressing model diversity in terms of aerosol loadings in Sec. 3.1 (see revised manuscript Page 5, L27).

- Page 7, L27: The CDN variability is primarily [typo in manuscript] related to anthropogenic sulfate, but this is not to be expected from the model parameterization of CDNC. CDNC in the models is a function of the mass/number concentration of hydrophilic aerosols, which is, in all the cases where it is specified in literature for the models used in this study, a linear sum of the relevant species, with no preferential weighting given to sulfate. CDN variability is primarily related to sulfate because sulfate accounts for most of the mass (as shown earlier in the manuscript, and in previous studies). The only model I can find where this finding might be a true result of the parameterization is HadGEM2-ES, where the number concentration of hydrophilic aerosols is a function only of sulfate and sea salt (Bellouin et al., 2007; Wilcox et al., 2015).

  Yes, you are right. Our statement was probably misleading. There is no special weighting of sulfate in the CDNC parameterization. Sulfate and other hydrophilic aerosols contribute to CDN. We rephrased the sentence (see revised manuscript Page 7 L28).

- Figure 3: Should the color bar be CDNC, not albedo?

  Figures 3 and 4 have been swapped, Figure 3 shows the CDN anomaly in the cloud fraction- albedo space.

- Figure 3/Page 8, L2: I think it would be interesting to see the results for LWP. We know from the parameterization in the models that increased CDN and LWP will increase albedo. It would be interesting to see which has the greatest effect in each model.

  You can see in Figure R5 the LWP anomaly in the albedo-cloud fraction space for the model NorESM in the Californian region as an example. For all models in all regions and experiments, the LWP anomaly and CDN anomaly show positive gradients. We decided to not include this figure in the manuscript and rather refer to Bender et al. (2016) where LWP-anomaly gradients are shown for models and satellite observations (see Page 8, L2 in the revised manuscript). Figure 7 in the manuscript shows how changes in CDN and LWP relate to changes in cloud albedo, across models. We added the LWP-gradient strength to Figure 7, i.e. to see if models with strong LWP-gradients also have large changes in cloud albedo. In general, the CDN-gradient is more positively correlated with change in cloud albedo than LWP-gradient.

- Figure 4: Should the color bar be AOD, not CDNC? Are Figures 3 and 4 paired with the wrong captions?

  Yes, Figures 3 and 4 have been swapped. Now Figure 3 shows the CDN anomaly in the albedo-cloud fraction space and Figure 4 shows the AOD anomaly.

- Figure 5: Is there a reason for using green twice? More distinct colors may be helpful.

  We tried to use colorblind friendly colors by using similar but graded colors (color scheme from colorbrewer2.org), to avoid having red and green in one figure. We changed the colors now in Figures 2 and 5.

- Page 8, L16: Why do you think this is?

  Absorbing aerosols above the cloud layer can reduce the scene albedo by absorbing solar radiation. This could lead to a reversed gradient. But as you can see in Sec. 3.4., we have not found overlying aerosols in the Namibian and Peruvian region in our set of models.

- Page 9, L14: Why do you think this is? LWP?

  Looking at the CDNC at cloud-top, MIROC shows that a cloud brightening on the month-to-month scale is mainly realized through changes in LWP rather than modifactions of the effective radius. These relations are also consistent with the LWP changes, that are smaller for MIROC than for MRI and NorESM.

- Page 13, L21: CDNC typically depends on other species, as well as sulfate. I think you're seeing the effect of having a larger mass concentration of sulfate aerosols.

  Yes, you are right. CDNC depends not only on sulfate aerosols. We rephrased the sentence (see Page 13, L34 in the revised manuscript).

Note: Figure 2 has been updated to correct for a slight shift in the geographical areas for two of the regions.

[revised manuscript text omitted]